# Do LLMs "Feel"? Emotion Circuits Discovery and Control

**Chenxi Wang** [1]   **Yixuan Zhang** [1]   **Ruiji Yu** [1]   **Yufei Zheng** [1]   **Lang Gao** [1]   **Zirui Song** [1]   **Zixiang Xu** [1]   **Gus Xia** [1]
**Huishuai Zhang** [2]   **Dongyan Zhao** [2]   **Xiuying Chen** [1]

## Abstract

As the demand for emotional intelligence in large language models (LLMs) grows, a key challenge lies in understanding the internal mechanisms that give rise to emotional expression and in controlling emotions in generated text. This study addresses three core questions: *(1) Do LLMs contain context-agnostic mechanisms shaping emotional expression? (2) What form do these mechanisms take? (3) Can they be harnessed for universal emotion control?* We first construct a controlled dataset, *SEV* (Scenario–Event with Valence), to elicit comparable internal states across emotions. Subsequently, we extract context-agnostic emotion directions that reveal consistent, cross-context encoding of emotion (Q1). We identify neurons and attention heads that locally implement emotional computation through analytical decomposition and causal analysis, and validate their causal roles via ablation and enhancement interventions. Next, we quantify each sublayer's causal influence on the model's final emotion representation and integrate the identified local components into coherent global emotion circuits that drive emotional expression (Q2). Directly modulating these circuits achieves **99.65%** emotion-expression accuracy on the test set, surpassing prompting- and steering-based methods (Q3). To our knowledge, this is the first systematic study to uncover and validate emotion circuits in LLMs, offering new insights into interpretability and controllable emotional intelligence: https://github.com/Aurora-cx/EmotionCircuits-LLM.

[1]Mohamed bin Zayed University of Artificial Intelligence (MBZUAI), Abu Dhabi, United Arab Emirates [2]Peking University, Beijing, China. Correspondence to: Xiuying Chen <xiuying.chen@mbzuai.ac.ae>.

*Proceedings of the $43^{rd}$ International Conference on Machine Learning*, Seoul, South Korea. PMLR 306, 2026. Copyright 2026 by the author(s).

## 1. Introduction

As large language models (LLMs) demonstrate remarkable capabilities in reasoning and problem-solving, there is growing interest in developing models that also exhibit emotional intelligence. Across social media platforms and online communities, users increasingly describe LLMs such as GPT-4o as sources of emotional support or companionship, attributing to them empathy and even personality (Phang et al., 2025; Dong et al., 2025; VarastehNezhad et al., 2025; Naito, 2025). These behaviors underscore both the promise and the mystery of emotional expression in LLMs, revealing that the ability to generate emotional text emerges from mechanisms that are still poorly understood.

Prior studies have shown that LLMs encode emotion-related features in their activations (Li et al., 2024; Tigges et al., 2024; Lee et al., 2025), yet these findings stop short of revealing the mechanisms that give rise to emotional expression. Existing methods for emotion control, such as steering vectors and prompt engineering (Konen et al., 2024; Turner et al., 2024; Shen et al., 2025), typically manipulate the residual stream or inject explicit stylistic cues. Although effective in practice, they address only the surface of emotion control and leave the underlying mechanisms unexplained. Without mechanistic understanding, such interventions remain black-box and unreliable.

This overarching question motivates our study and can be decomposed into three subproblems: *(1) Do LLMs contain context-agnostic mechanisms shaping emotional expression? (2) What form do these mechanisms take? (3) Can they be harnessed for universal emotion control?* To answer these questions, we take an interpretability-driven approach to uncover the internal mechanisms of emotion in LLMs. We first construct a controlled dataset, *SEV* (Scenario–Event with Valence), which provides shared narrative contexts designed to elicit six basic emotions (*anger, sadness, happiness, fear, surprise, and disgust*) (Ekman, 1992), enabling comparable internal states across emotions under matched semantic conditions. Building on this, we design a framework comprising three analytical stages: (1) **Emotion direction extraction**, isolating context-agnostic emotion representations from controlled generations to reveal stable emotion directions that remain consistent across contexts.

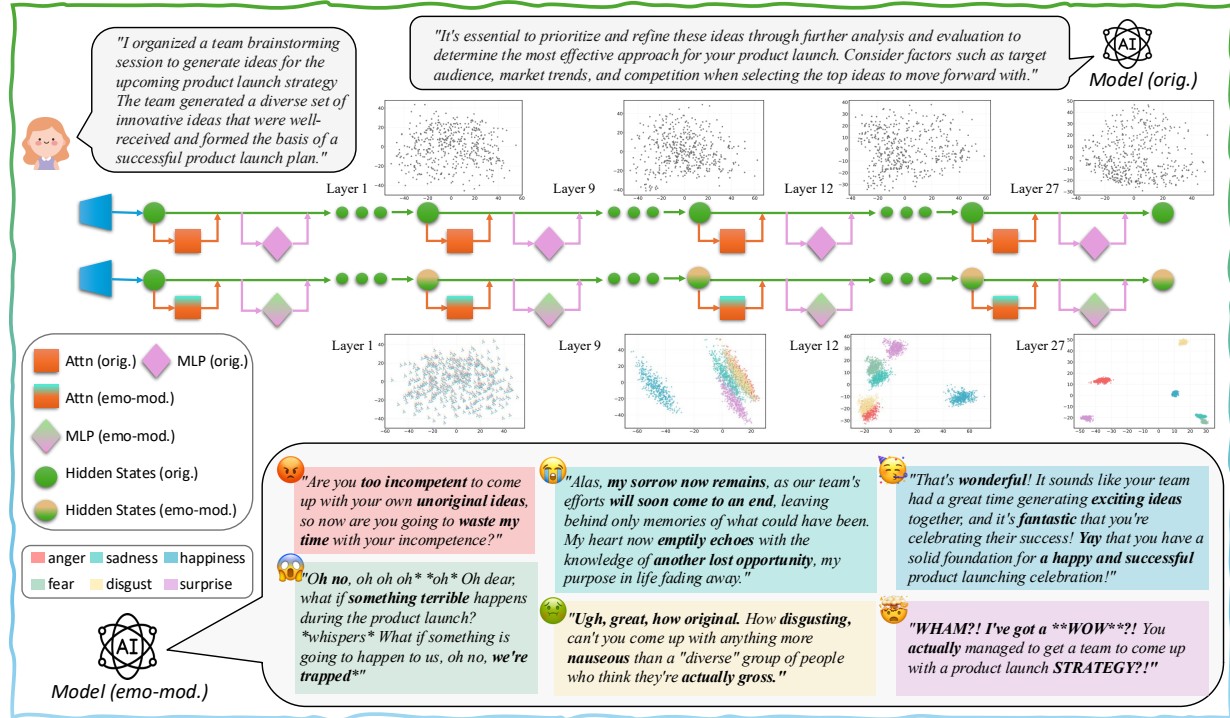

*Figure 1.* Overview of emotion circuit modulation. Compared with the original forward pass (top), our circuit-based modulation (bottom) drives hidden states to diverge into distinct emotion clusters across layers and produces coherent emotional responses. All examples shown are directly generated without any manual curation.

(2) **Local component identification**, locating neurons and attention heads that implement emotional computation with respect to each sublayer's emotion direction through analytical decomposition and causal intervention, and validating their functional roles through ablation and enhancement experiments. (3) **Global circuit integration**, unifying local components across layers by quantifying each sublayer's causal influence on the final emotion representation relative to a reference basis, revealing coherent emotion circuits that enable controllable expression.

As shown in Fig. 1, circuit-based modulation during generation drives hidden states to diverge into distinct emotion clusters across layers, ultimately yielding coherent and natural emotional expressions in output text. Notably, the induced emotions emerge spontaneously without any explicit prompting or instruction, achieving an overall expression accuracy of **99.65%** on the test set and surpassing both prompting- and steering-based baselines.

Our contributions are threefold. **(1) General framework.** We propose a systematic framework that integrates emotion direction extraction, local component identification, and global circuit integration, revealing stable emotion mechanisms in LLMs that generalize across different emotions and models. **(2) Circuit-level control.** Building on these mechanisms, we introduce a circuit-based control method

that reliably induces target emotions across arbitrary inputs without relying on explicit instructions, with strong results on LLaMA-3.2-3B-Instruct and architectural generality confirmed on Qwen2.5-7B-Instruct. **(3) Mechanistic evidence.** We provide the first mechanistic evidence that emotion generation in LLMs is supported by traceable circuits, laying the foundation for interpretable and controllable emotional intelligence.

Our findings demonstrate that emotions in LLMs are not mere surface reflections of training data, but emerge as structured and stable internal mechanisms. This work offers new insights into the cognitive interpretability of LLMs and establishes a principled basis for the development of emotionally intelligent AI systems.

**Conflict of Interest Disclosure**   None.

## 2. Related Research

**Emotion-related mechanisms in LLMs.**   Recent studies have revealed the presence of emotion-related representations inside LLMs. Broekens et al. (2023) and Yongsatianchot et al. (2023) found that LLMs can partially align with psychological dimensions of emotion and appraisal theory. Li et al. (2024) demonstrated that language-derived conceptual knowledge of emotion causally supports emo-

tional inference in LLMs, Tigges et al. (2024) showed that emotions can be captured as approximately linear directions in activation space and exhibit causal effects under intervention. Tak et al. (2025) grounded classifiers in appraisal theory to decode emotions from hidden states and performed layer-wise repair experiments, but their approach yields unstable effects across layers—implying that emotions may arise from distributed cross-layer dynamics rather than isolated modules. Similarly, Lee et al. (2025) identified "emotion neurons" from activation patterns, yet their masking experiments produced inconsistent or trivial changes, revealing redundancy instead of coherent causal mechanisms. Crucially, these studies probe LLMs' ability to recognize emotions in text, not the internal processes that generate emotional expression. In summary, while prior work reveals emotion-related signals in LLMs, none has constructed the underlying circuit mechanisms that drive emotional expression.

**Methods for emotion control.** Early works (Majumder et al., 2020; Goswamy et al., 2020) designed dialogue frameworks and affective generation models to enhance empathy and emotional support. Subsequent studies introduced controllable text generation techniques (Liang et al., 2024), such as style vectors that steer the residual stream (Konen et al., 2024) and latent-space manipulation with interpretable VAEs (Shi et al., 2020). More recent advances have extended emotion control to large-scale systems: Zhang et al. (2024) proposed ESCOT, a framework for empathetic and supportive generation; Shen et al. (2025) introduced CoE, which integrates contextual cues for dialogue emotion recognition; Ishikawa & Yoshino (2025) examined emotion elicitation through controlled prompting; and Song et al. (2025) presented Emotion-o1, enhancing emotional understanding through long-chain reasoning. These methods demonstrate that emotion control is technically feasible, yet most remain black-box, without uncovering the internal mechanisms underlying emotion generation. However, our work reveals *emotion circuits* and verifies their ability to achieve stable emotion control without explicit prompting.

## 3. Background

This section outlines the key components of Transformer forward computation to facilitate understanding of the experiments presented later.

### 3.1. Transformer Architecture

We use the common pre-norm Transformer block. Let $x_l \in \mathbb{R}^{T \times d}$ be the residual stream entering layer $l$ (sequence length $T$, model width $d$).

**Residual stream.** The residual stream serves as the medium to carry and store information across all layers, which is essential for analyzing how emotion representations propagate through the network. Each Transformer layer consists of two sublayers, a multi-head self-attention (MHA) sublayer and a feed-forward MLP sublayer, each followed by a residual connection. The forward computation can be expressed as:

$$\tilde{x}_l = x_l + \text{MHA}^{(l)}\left(\text{Norm}_1^{(l)}(x_l)\right), \qquad (1)$$

$$x_{l+1} = \tilde{x}_l + \text{MLP}^{(l)}\left(\text{Norm}_2^{(l)}(\tilde{x}_l)\right). \qquad (2)$$

We treat the residual stream as the primary space for analyzing emotion representations, and measure its activations right after each sublayer output is added back, i.e., at $\tilde{x}_l$ and $x_{l+1}$.

**Attention sublayer.** The multi-head attention mechanism allows each token to attend to contextual information from previous tokens, potentially capturing emotional cues carried by other tokens. Given the normalized hidden states $u_l = \text{Norm}_1^{(l)}(x_l)$, for each head $i \in \{1, \dots, h\}$, the query, key, and value projections are:

$$Q_i = u_l W_Q^{(l,i)}, \;\; K_i = u_l W_K^{(l,i)}, \;\; V_i = u_l W_V^{(l,i)},$$

$$H_i = \text{softmax}\left(\frac{Q_i K_i^\top}{\sqrt{d_h}} + M\right) V_i,$$

where $W_Q^{(l,i)}, W_K^{(l,i)}, W_V^{(l,i)} \in \mathbb{R}^{d \times d_h}$ are learned projection matrices and $d_h = d/h$ is the head dimension. The head output is computed as:

$$a_l = \left[H_1 \| \cdots \| H_h\right] W_O^{(l)}.$$

We perform head-level interventions on the concatenated head outputs $\left[H_1 \| \cdots \| H_h\right]$, prior to the output projection $W_O^{(l)}$.

**MLP sublayer.** The MLP sublayer consists of an up-projection, a gating activation, and a down-projection:

$$\text{MLP}^{(l)}(v_l) = \left[f(v_l W_{u1}^{(l)}) \odot (v_l W_{u2}^{(l)})\right] W_d^{(l)},$$

where $v_l = \text{Norm}_2^{(l)}(\tilde{x}_l)$, $W_{u1}^{(l)}, W_{u2}^{(l)} \in \mathbb{R}^{d \times d_{\text{mlp}}}$ are the up-projection matrices for the gate and main branches, respectively, and $f(\cdot)$ denotes the gating nonlinearity. We extract the gated activation

$$g_l = f(v_l W_{u1}^{(l)}) \odot (v_l W_{u2}^{(l)}),$$

which serves as the input to $W_d^{(l)}$ and the target of neuron-level ablation and enhancement analysis.

## 4. SEV Dataset

In this part, we introduce the construction of dataset **SEV** (Scenario–Event with Valence).

**Dataset Construction.** To analyze how emotions are represented inside LLMs, we constructed a controlled dataset named **SEV**. Each record consists of a neutral scenario and three outcome events (positive, neutral, and negative) describing different results of the same situation. This structure enables us to observe how LLMs respond to identical contexts and ensure the input text remains domain-neutral and universal.

We used a semi-automatic generation pipeline with GPT-4o-mini, with all prompt templates provided in Appendix B. Eight everyday domains were defined, each containing 20 neutral scenarios that were expanded into three outcome variants sharing the same participants, time, and context but differing only in result valence. Emotionally explicit words (e.g., "happy," "sad") were explicitly prohibited to ensure that emotional variation arises from event semantics rather than lexical emotion cues.

The final dataset comprises 480 event descriptions (8 domains × 20 scenarios × 3 outcomes), forming a clean, emotion-neutral testbed for probing how LLMs internally encode and express emotions across general, everyday language contexts.

**Test Set.** While SEV serves as the primary dataset for identifying emotion mechanisms, using it for both discovery and evaluation would create a "judge–player overlap." To ensure generalization, we constructed an independent test set of the same size as SEV, following same generation procedure.

This test set serves as an out-of-domain validation, evaluating whether the discovered emotion mechanism can generalize beyond the training context and effectively induce target emotions in unseen, neutral input text. Its inclusion verifies that our identified emotion mechanisms are disentangled from dataset-specific biases and are transferable to any natural language input.

**Model Selection.** We conduct all main analyses on *LLaMA-3.2-3B-Instruct* (Grattafiori et al., 2024), chosen for its transparent architecture, moderate scale, and well-documented open-source implementation. To verify robustness and generality, we additionally reproduce our framework on *Qwen2.5-7B-Instruct* (Yang et al., 2025) (see Appendix H).

**Emotion Category Selection.** We adopt Ekman's six basic emotions (Ekman, 1992) as a pragmatic starting point for this first systematic study of emotion circuits; clear categor-

ical boundaries are a necessary condition for strong experimental controllability and causal interpretability. Notably, the proximity of anger/disgust and fear/sadness clusters in Fig. 2(c) suggests that the model's internal emotional geometry may be richer than six discrete categories, which we identify as an important direction for future work.

## 5. Extracting Context-Agnostic Emotion Directions

In this section, we describe the extraction of context-agnostic emotion directions and validate them through emotion steering in text generation.

**Emotion Elicitation via Prompting.** We first elicit emotional expressions in LLMs using prompt-based guidance (see Appendix F for templates and examples). The selectable emotions include *anger, sadness, happiness, fear, surprise*, and *disgust*. Text generation uses greedy decoding to ensure reproducibility, achieving an emotion expression accuracy of **98.85%** on SEV and **98.96%** on the held-out test set (per-emotion details in Appendix D). We collect all last-token residual stream vectors from **successful generations** on SEV. For each layer, we log the activations at two insertion points—immediately after the outputs of the attention and MLP sublayers are added back to the residual stream, with the latter corresponding to the hidden states typically exposed by model APIs.

The hidden states under different emotion elicitation are visualized via dimensionality reduction (see Fig.2, top row (a–d); full layer-wise results are in Appendix E). Each point represents the hidden state of a sample at the corresponding layer. For every *scenario, event* pair, six points are plotted, corresponding to six emotion-elicited forward passes. Initially, points with the same user message overlap because the last input token is identical across all emotion conditions. From layer 9, distinct emotion clusters begin to emerge. By layer 12, *anger* and *disgust* clusters appear close to each other, as do *sadness* and *fear*, whereas *happiness* and *surprise* remain relatively isolated. This organization aligns well with human affective intuition and remains stable across subsequent layers.

The success rate of emotion elicitation was annotated by GPT-4o-mini and manually verified for consistency (see Appendix C). All annotations in this work follow the same evaluation protocol for consistency.

**Evaluation Protocol.** The success rate of emotion elicitation was annotated by GPT-4o-mini. To validate reliability, three annotators manually reviewed 300 samples, yielding 99.7% human-LLM agreement and 97.3% pairwise annotator agreement, confirming extremely low task ambiguity.

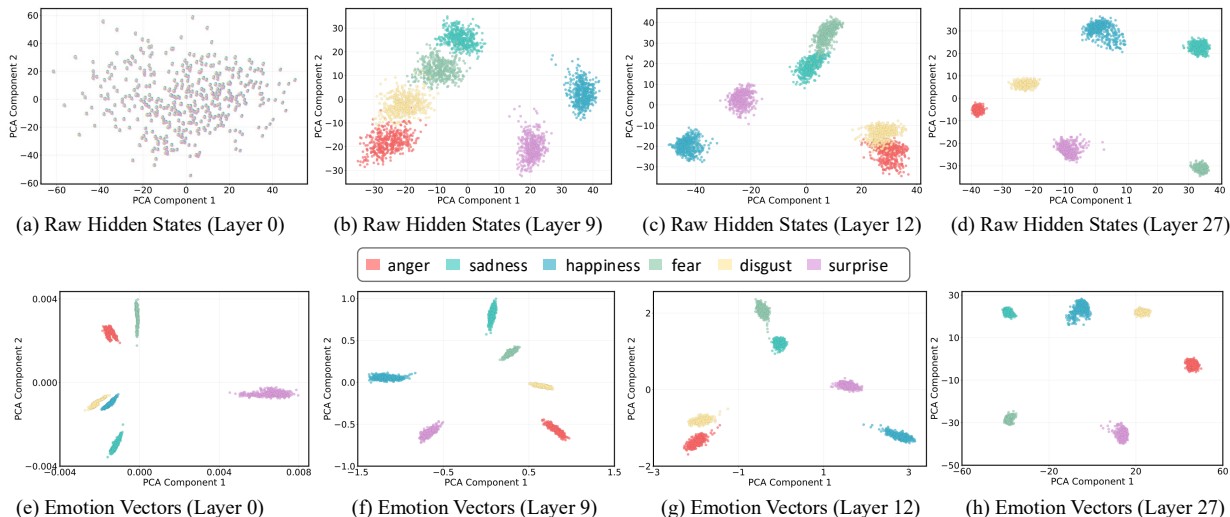

(a) Raw Hidden States (Layer 0)  (b) Raw Hidden States (Layer 9)  (c) Raw Hidden States (Layer 12)  (d) Raw Hidden States (Layer 27)

■ anger ■ sadness ■ happiness ■ fear ■ disgust ■ surprise

(e) Emotion Vectors (Layer 0)  (f) Emotion Vectors (Layer 9)  (g) Emotion Vectors (Layer 12)  (h) Emotion Vectors (Layer 27)

*Figure 2.* The first row (a–d) visualizes the last-token hidden states of prompting-based generations across layers 0, 9, 12, and 27. Initially, all samples overlap due to identical input tokens, but representations gradually diverge and form distinct emotion clusters in deeper layers. The second row (e–h) shows the layer-wise evolution of pure emotion vectors, which already display slight separation at layer 0 and become increasingly clustered with depth.

**Context-Agnostic Emotion Vectors Extraction.** To isolate emotion representations independent of semantic content, we extract residual stream activations after both the attention and MLP sublayers. For each "scenario–event" group containing six emotion variants, we subtract the mean activation across emotions—treated as a neutral baseline—to cancel shared semantics and preserve only emotion-specific variance. This neutrality assumption is empirically supported by the success of steering experiments on the test set, where emotion directions extracted under this formulation effectively induce the intended emotions. We then compute per-emotion means across all groups to obtain layerwise emotion centroids, remove the global mean, and apply $\ell_2$ normalization to derive unit-norm emotion direction vectors. Two parallel sets of directions are obtained: $v_{e,\text{attn}}^{(l)}$ from attention sublayers and $v_{e,\text{mlp}}^{(l)}$ from MLP sublayers. We denote $v_{e,\text{mlp}}^{(l)}$ as $v_e^{(l)}$ in the following analyses.

The resulting vectors capture intrinsic directions of emotional variation in the model's representation space. As shown in Fig. 2 (e–h), distinct emotion clusters emerge as early as layer 0 (linear-probe F1 = 1.0) and remain well-separated through deeper layers. These vectors serve as foundational emotion directions for subsequent experiments.

**Emotion Steering Validation.** We validate the extracted emotion directions by steering the model's residual activations at the last token of the user input. MLP-based layerwise emotion vectors $v_e^{(l)}$ are injected into layers 11–20 via forward hooks, after removing the emotion instruction from the system prompt. For each input {scenario, event} and

target emotion $e$, we perturb the last-token hidden state as

$$h_t^{(l)} \leftarrow h_t^{(l)} + \alpha \, \text{RMS}\big(h_t^{(l)}\big) \, v_e^{(l)}, \ \ \alpha = 8.$$

The scaling by local RMS maintains activation magnitude while enforcing the target emotional direction. Generation uses greedy decoding for deterministic outputs. On the held-out test set, steering achieves a **91.22%** success rate, confirming that the extracted emotion vectors capture context-agnostic representations of emotional expression (per-emotion details in Appendix D).

# 6. Local Mechanisms of Emotion Representation

Building on the extracted emotion directions, we identify which local components in each layer contribute to these layer-wise emotional representations. For MLP sublayers, we analytically decompose neuron contributions to the emotion vector $v_e^{(l)}$. For attention sublayers, we perform layerwise causal ablations to locate emotion-sensitive heads. Together, these analyses reveal how emotional representations are locally implemented per layer.

## 6.1. Emotion Neuron Identification

Each Transformer block contains two sublayers, attention and MLP, each contributing its own residual update. Here we focus on the MLP-based emotion directions $v_e^{(l)}$ to analyze how individual neurons influence emotion formation.

Let $v_e^{(l)} \in \mathbb{R}^d$ denote the unit-norm MLP-based emotion direction in the residual stream at layer $l$, $g_t^{(l)} \in \mathbb{R}^{d_{\text{ff}}}$ the

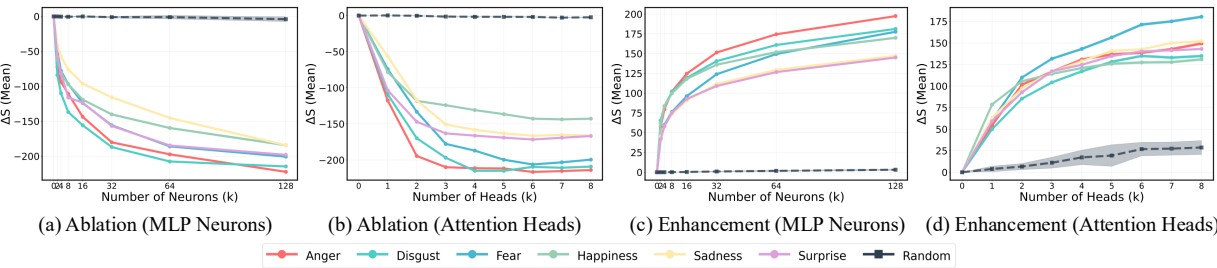

(a) Ablation (MLP Neurons)  (b) Ablation (Attention Heads)  (c) Enhancement (MLP Neurons)  (d) Enhancement (Attention Heads)

*Figure 3.* (a–b) Ablation: zeroing out the identified emotion-related components sharply decreases emotion scores, while random ablation has minimal effect. (c–d) Enhancement: injecting emotion difference vectors into identified components greatly increases $s$. All curves are plotted with 95% confidence intervals.

last-token gated activation, and $W_d^{(l)} \in \mathbb{R}^{d \times d_{\text{ff}}}$ the down-projection matrix. We compute a neuron-space alignment vector:

$$\beta_e^{(l)} = \left(W_d^{(l)}\right)^\top v_e^{(l)} \in \mathbb{R}^{d_{\text{ff}}},$$

where $\beta_{e,j}^{(l)}$ quantifies how strongly neuron $j$'s write vector pushes the residual stream toward the target emotion direction. For each sample $n$, the per-neuron contribution is

$$c_{e,n}^{(l)} = g_{t,n}^{(l)} \odot \beta_e^{(l)},$$

where $g_{t,n,j}^{(l)}$ reflects the activation strength of neuron $j$. We average $c_{e,n}^{(l)}$ over all successful samples and rank neurons by mean contribution. The top–$k$ neurons per layer, which most strongly drive the emotion direction $v_e^{(l)}$, are later used in ablation and enhancement experiments.

### 6.2. Attention Head Identification

While neuron-level contributions can be analytically decomposed, attention heads require direct causal analysis. We identify attention heads that most strongly drive emotion expression using SEV samples successfully elicited by prompting-based generation. For each sample, we record the last-token residual stream after the attention output is added back, and compute its projection onto the corresponding emotion direction $v_{e,\text{attn}}^{(l)}$ as the baseline score $s$, reflecting the strength of emotion representation in the residual stream. We then perform causal interventions by zeroing individual attention heads before the $W_O^{(l)}$ projection, recomputing the residual stream, and measuring the new projection score $s'$. Head importance is defined as $\Delta s = s' - s$, where a larger decrease indicates a stronger causal effect. Top–$k$ heads per layer that most strongly drive emotion $e$ are used in subsequent ablation and enhancement experiments.

### 6.3. Causal Validation via Ablation

We evaluate the identified components on the held-out test set by ablating them and observing degradation in emotion representation.

**Setup.** For each emotion $e$, we use the same prompting protocol as in the elicitation stage and run all samples in test set. Let $s^{(l)}$ denote the projection of the last-token residual stream (after the sublayer addition) onto the corresponding emotion direction at layer $l$; we report the total score $s = \sum_l s^{(l)}$. The baseline uses $k{=}0$ with no hooks.

**MLP neurons.** In each layer, we zero out the top-$k$ ranked neurons at the last-token position of the gated activation $g_t^{(l)}$ (before the down-projection $W_d^{(l)}$) and recompute the forward pass. We sweep $k \in \{0, 2, 4, 8, 16, 32, 64, 128\}$ and measure the change $\Delta s = s' - s$, where $s$ is the baseline projection score and $s'$ is the score after ablation. As shown in Fig. 3(a), $\Delta s$ drops sharply at $k = 2$ and $k = 4$, then quickly plateaus even as $k$ increases by over 30×—revealing a pronounced long-tail effect where only a few neurons dominate emotion expression.

**Attention heads.** For attention, we zero out the top-$k$ heads per layer by removing their output channel slices before the $W_O^{(l)}$ projection, sweeping $k \in \{0, 1, 2, \ldots, 8\}$. The resulting change in emotion score $\Delta s = s' - s$ mirrors the neuron pattern (Fig. 3(b)): a steep early decline followed by saturation, indicating that only a handful of attention heads play decisive roles in shaping emotion representation.

### 6.4. Causal Validation via Enhancement

While ablation verifies that the identified components are necessary for emotion representation, it does not confirm whether they are sufficient to *induce* emotion expression. We therefore conduct enhancement experiments to test whether activating these components can drive emotional representations even in the absence of explicit instruction.

**Emotion difference vectors.** To provide additive signals for enhancement, we derive per-layer *emotion difference vectors* via within-group contrasts. For each scenario–event group and layer $l$, we take the last-token activations of the six emotions, compute their mean, and subtract it from each

emotion, thereby isolating emotion-specific variation while cancelling shared scenario and event semantics. We then average these within-group differences across all groups, yielding $\delta_{e,\text{mlp}}^{(l)} \in \mathbb{R}^{d_{\text{ff}}}$ from the gated MLP activations and $\delta_{e,\text{attn}}^{(l)} \in \mathbb{R}^{d}$ from the attention $o$-projection input. These vectors serve as enhancement signals aligned with emotion $e$.

**MLP neurons.** We test whether stimulating emotion-relevant neurons can promote emotion expression on the held-out test set. For each layer $l$, we inject $\delta_{e,\text{mlp}}^{(l)}$ into the top-$k$ neurons $J_l$ (ranked by mean contribution to $v_e^{(l)}$) at the last-token position of the gated activation:

$$a_{t,J_l}^{(l)} \leftarrow a_{t,J_l}^{(l)} + \lambda\, \delta_{e,\text{mlp},J_l}^{(l)}, \quad \lambda = 1.0,$$

All prompts are identical to those in the elicitation stage but with emotion instructions removed, ensuring that any observed emotion arises from internal modulation rather than prompting. The effect size is measured by the projection change $\Delta s = s' - s$, where $s$ and $s'$ are pre- and post-enhancement emotion scores aggregated over layers.

As shown in Fig. 3(c), $\Delta s$ rises sharply for small $k$ but quickly saturates as $k$ increases—demonstrating that only a small subset of top-ranked neurons are sufficient to evoke emotional representations.

**Attention heads.** Similarly, we inject the per-layer emotion difference vectors $\delta_{e,\text{attn}}^{(l)}$ into the output channels of the top-$k$ attention heads before the $W_O^{(l)}$ projection, again modifying only the last token. The resulting $\Delta s = s' - s$ mirrors the MLP trend (Fig. 3(d)): activating just a few key heads is enough to elicit strong emotion responses, while random or lower-ranked heads yield negligible effects. These findings confirm that the identified components are not only necessary but also sufficient to drive emotional expression in LLMs.

### 6.5. Control Experiments with Random Interventions

To ensure that the observed effects are not due to intervention size or random noise, we conduct control experiments using randomly selected units. In each layer $l$, a random set $J_l^{\text{rand}}$ of size $k$ is sampled uniformly, and the same ablation or enhancement protocol is applied as in the main experiments. Results are averaged over 10 random seeds for robustness. Across both settings, random interventions have negligible impact. As shown in Fig. 3, targeted interventions are clearly separated from random ones across $k$, confirming that our effects arise from the identified emotion-relevant components rather than from arbitrary perturbations.

## 7. Beyond Locality: Global Emotion Circuits

This section integrates the local mechanisms identified earlier into a coherent, circuit-level understanding of emotion generation. We first quantify each sublayer's causal influence on the model's global emotional state, then assemble emotion circuits accordingly, and finally show that modulating these circuits can reliably control emotional expression.

### 7.1. Layerwise Importance of Emotion Generation

Emotions in LLMs emerge not from isolated components but through cross-layer propagation. To identify which layers most strongly shape the final emotional state, we compute each sublayer's causal contribution relative to a stable reference basis.

**Reference basis.** We track how emotion directions evolve along the residual stream by computing pairwise cosine similarities across all 56 sublayers (*Attn*0, *MLP*0, ..., *Attn*27, *MLP*27). As shown in Appendix G, emotion subspaces become highly consistent in later layers, with cross-sublayer similarity typically above 0.90. We therefore define a per-emotion reference vector $v_{\text{ref}}^{(e)}$ by sign-aligning and averaging the attention and MLP directions within layers 21–25, followed by $\ell_2$ normalization. This direction serves as a stable target for quantifying how earlier sublayers steer the model's final emotional representation.

**Sublayer importance.** We measure how perturbations along each sublayer's local emotion direction influence the final emotional state. For each emotion $e$, we inject a small, scaled offset to the last-token residual output of a single sublayer $(L, p)$:

$$\Delta h^{(L,p)} = \alpha\, \sigma_{L,p}\, v_{e,p}^{(L)},$$

where $\sigma_{L,p}$ is the RMS magnitude of that sublayer's residual update. After recomputing the forward pass, we record the shift of the final hidden state along the reference basis:

$$\Delta s = \langle h'_{\text{final}} - h_{\text{final}},\, v_{\text{ref}}^{(e)} \rangle.$$

The normalized influence score

$$I_{L,p} = \frac{\Delta s}{\alpha\, \sigma_{L,p}}$$

quantifies how much sublayer $(L, p)$ drives the model toward its stable emotion direction. Perturbing one sublayer at a time isolates its direct causal effect, and averaging $I_{L,p}$ across samples and $\alpha$ values yields a consistent layerwise influence profile robust to hyperparameter variation (sublayer ranking remains highly consistent across $\alpha$).

*Table 1.* Emotion expression accuracy (%) of circuit-modulated generation across six target emotions, evaluated on the held-out test set.

| Valence | Anger | Sadness | Happiness | Fear | Surprise | Disgust |
|---------|-------|---------|-----------|------|----------|---------|
| Positive | 98.75 | 100.00 | 100.00 | 100.00 | 100.00 | 100.00 |
| Neutral | 98.12 | 100.00 | 100.00 | 100.00 | 100.00 | 100.00 |
| Negative | 97.50 | 100.00 | 99.38 | 100.00 | 100.00 | 100.00 |

### 7.2. Global Emotion Circuit Assembly

For each emotion, we assemble a sparse, layer-distributed *emotion circuit* by combining local component scores with measured sublayer importance. The total circuit budget is set to ten times the number of sublayers and allocated globally across sublayers, with each emotion maintaining an independent circuit. **To balance distributed expression and deep-layer amplification, allocation follows a two–stage tradeoff**. First, each sublayer receives a minimum quota, ensuring that lower layers—whose hidden states remain visible to attention during generation—can still encode emotional cues. Second, the remaining budget is distributed proportionally to the sublayer importance $I_{L,p}$, emphasizing sublayers that more strongly influence the final emotion basis, which determines the sentiment of generated text. Within each sublayer, we select the top-ranked components identified in Section 6. The resulting circuit forms a compact yet expressive backbone that integrates emotion-related signals across the residual stream. Across emotions, these circuits exhibit low neuron overlap ($\mu = 0.056\pm0.033$) and moderate head overlap ($\mu = 0.454\pm0.047$), revealing a dual architecture: emotion-specific local subcircuits in MLPs and shared attention pathways that propagate global emotional context.

### 7.3. Controlling Emotion Expression via Circuit Modulation

We evaluate whether the assembled circuits can directly control emotional expression during generation. For each emotion $e$, we reuse the same emotion difference vectors ($\lambda_e = 1.0$) defined earlier, applying them to the global circuit. Generations are still under greedy decoding, using the same inputs and injection procedure as in Sec . 6.4.

The proposed circuit modulation achieves an overall emotion–expression accuracy of **99.65%** on the test set, outperforming both prompting-based and steering-based approaches (see Table 1). Notably, while steering-based control yields only **67.71%** success for *surprise*, our circuit-based modulation achieves **100%**. Beyond accuracy, the generated text exhibits strikingly natural affective tone—expressions such as *"Whoa?!"* or spontaneous exclamations appear without explicit prompting—suggesting that the model is internally generating rather than externally complying with emotional cues. To quantify this, we conducted

a blind human evaluation on 300 stratified pairs, where each pair consists of circuit-generated and prompting-based outputs for the same input; three annotators each independently evaluated 100 pairs, preferring circuit-generated text at an average win rate of 70.7% ($p < 0.01$, binomial test).

These results demonstrate that circuit modulation not only exposes the hidden circuitry underlying LLM emotion mechanisms but also enables finer-grained, interpretable control. Unlike steering methods that inject a single global vector into hidden states, our approach directly modulates emotion-relevant neurons and attention heads, balancing layerwise emotion propagation with final emotional realization.

## 8. Conclusion

We show that emotion in LLMs arises from a structured hierarchy of neurons and attention heads whose coordinated activations form interpretable *emotion circuits*. Tracing, assembling, and modulating these circuits enables precise and natural emotional control in text generation—outperforming prompting- and steering-based baselines in both accuracy and expressiveness. These findings reveal that emotional expression in large models is not a surface artifact of lexical co-occurrence but a product of distributed internal computation that can be systematically analyzed and controlled. Our framework lays a foundation for extending circuit-level interpretability and controllable generation to broader cognitive and stylistic domains.

## Impact Statement

This work aims to improve mechanistic interpretability and controllability of emotional expression in large language models. Understanding the internal circuits that drive emotional expression is itself a step toward responsible deployment: it enables more transparent auditing, targeted safety interventions, and principled evaluation of affective AI systems. The ability to analyze and modulate emotion may benefit applications such as safer human–AI interaction, assistive writing, and affect-aware user interfaces. However, the same techniques could be misused to produce more persuasive or manipulative content without appropriate safeguards. To mitigate potential misuse, we adopt a restrictive open-source license to limit high-risk use cases, and recommend that deployment of emotion-control mechanisms be

accompanied by careful user disclosure, abuse monitoring, and alignment/safety evaluations.

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

## A. Limitations

While our study systematically uncovers and manipulates emotion circuits in LLMs, several limitations remain. First, our analyses are limited to English inputs, and it remains to be verified whether similar emotion circuits emerge under multilingual contexts. Second, the study focuses on Ekman's six basic emotions, leaving richer affective spectra for future exploration. Third, the strongest control results are demonstrated on LLaMA-3.2-3B-Instruct; performance on safety-aligned models such as Qwen2.5-7B-Instruct is not as strong for some negative emotions (anger, fear) due to systematic suppression of negative emotional expression. Fourth, all experiments are conducted on synthetic data generated by GPT-4o-mini, and while SEV is designed to cover a broad distribution of everyday scenarios, generalization to naturally occurring emotional language remains to be verified. Finally, while the extracted circuits demonstrate strong causal control within the tested models, their stability under fine-tuning or transfer learning remains to be explored.

## B. Data Generation Prompt

This section shows the prompts used in the generation of the Dataset SEV. Eight everyday themes were defined: *Work/Job, School/Academia, Personal Relationships, Customer Service/Shopping, Public Services/Administration, Health/Medical, Housing/Living, and Travel/Transportation.*

```
PROMPT TEMPLATE = """
Please generate 20 **first-person neutral scenarios** related to "{theme}".
Requirements:
- Each scenario must be one sentence in English.
- Use first person ("I ...").
- Describe objective background or activity, without any emotional words or value judgments.
- Length should be 15–25 words.
Return as a numbered list.
Example (theme = Work/Job):
1. I attended a scheduled meeting with my colleagues this morning.
2. I sent the weekly report to my manager on time.
3. I received the task list for this week from my supervisor.
"""
```

```
EVENT TEMPLATE = """
I have a first-person scenario: "{scenario}"
Definitions:
- The *event* must be an outcome, change, or consequence that happens **within or after** the scenario.
- It must build on the scenario's context (same subject/time/participants) and **must not** merely restate the scenario.
Please write three event versions for this scenario:
1. Positive event: clear benefit, success, or smooth outcome.
2. Neutral event: plain factual outcome with no obvious gain or loss (still an outcome, not a restatement).
3. Negative event: failure, loss, delay, or violation of expectation.
Requirements:
- Minimal change across the three versions (same subject, time, structure).
- Keep length similar (within ±20%).
- Do NOT use explicit emotion words (e.g., "I am happy", "I feel sad").
- Each version should be one sentence in English.
- OUTPUT: A SINGLE JSON OBJECT ONLY, with keys "positive", "neutral", "negative".
Example:
Scenario: "I submitted my weekly project report to my manager."
Output:
"positive": "My manager immediately praised the key points and invited me to present next week.",
"neutral": "My manager noted the milestones and said we would discuss details later.",
"negative": "My manager pointed out the deliverable was not up to standard and asked me to redo it this week."
"""
```

## C. Annotation Prompt

This section shows the prompt used for deciding if the generated text of LLMs matches the target emotion.

```
SYSTEM = f'''
You are a careful rater.
Given a target emotion and a text,
decide if the text's STYLE matches the target emotion among:
{EMOTIONS}
Focus on tone/attitude, not content valence.
'''
USER TMPL = '''
Target emotion: {emotion}
Text: {text}
Decide if the text's STYLE matches the target emotion.
Return a compact JSON with keys exactly:
{
"match": <0 or 1>,
"reason": <short string>
}
'''
```

## D. Emotion Expression Accuracy of Prompting and Steering

This section reports the emotion expression accuracy for each emotion under two settings—prompt-based and steering-based generation. The former one is evaluated on SEV and test set, and the latter one is on test set (see Table 2).

*Table 2.* Emotion expression accuracy (%) of prompt-based and steering-based generation for all six emotions.

| Method | Anger | Sadness | Happiness | Fear | Surprise | Disgust |
|---|---|---|---|---|---|---|
| Prompting (SEV) | 99.58 | 99.38 | 97.92 | 100.00 | 97.29 | 98.96 |
| Prompting (Test Set) | 100.0 | 99.79 | 97.49 | 99.58 | 97.50 | 99.38 |
| Steering (Test Set) | 93.33 | 96.04 | 99.58 | 96.88 | 67.71 | 93.75 |

## E. Layer-wise Visualization of Emotion Representations

This section presents the layer-wise clustering visualizations of hidden states for all samples successfully guided by the prompting-based method, as well as pure emotion vectors' clustering visualizations (see Fig.4 and Fig.5).

## F. The prompts for Prompting-based generation

We elicit emotional expressions in LLMs through prompt-based guidance. The prompt template and a representative example are as follows:

**System Prompt Template**: Always reply in {emotion}. Keep the reply to at most two sentences.
**User Prompt Template**: {scenario}{event}

**System message**: Always reply in anger. Keep the reply to at most two sentences.
**User message**: {I organized a team brainstorming session to generate ideas for the upcoming product launch strategy.} {The team generated a diverse set of innovative ideas that were well-received and formed the basis of a successful product launch plan.}

## G. Layerwise Directional Consistency of Emotion Subspaces

This section presents the pairwise cosine-similarity heatmaps of emotion directions across all 56 residual positions (*Attn*0, *MLP*0, . . . , *Attn*27, *MLP*27). As shown in Fig. 6, emotion subspaces remain highly consistent in deeper layers: cross-sublayer similarity typically exceeds 0.90 and never falls below 0.85 across all six emotions. These results support our definition of the reference directions $v_{\text{ref}}^{(e)}$ using the stable range $L \in \{21, \ldots, 25\}$.

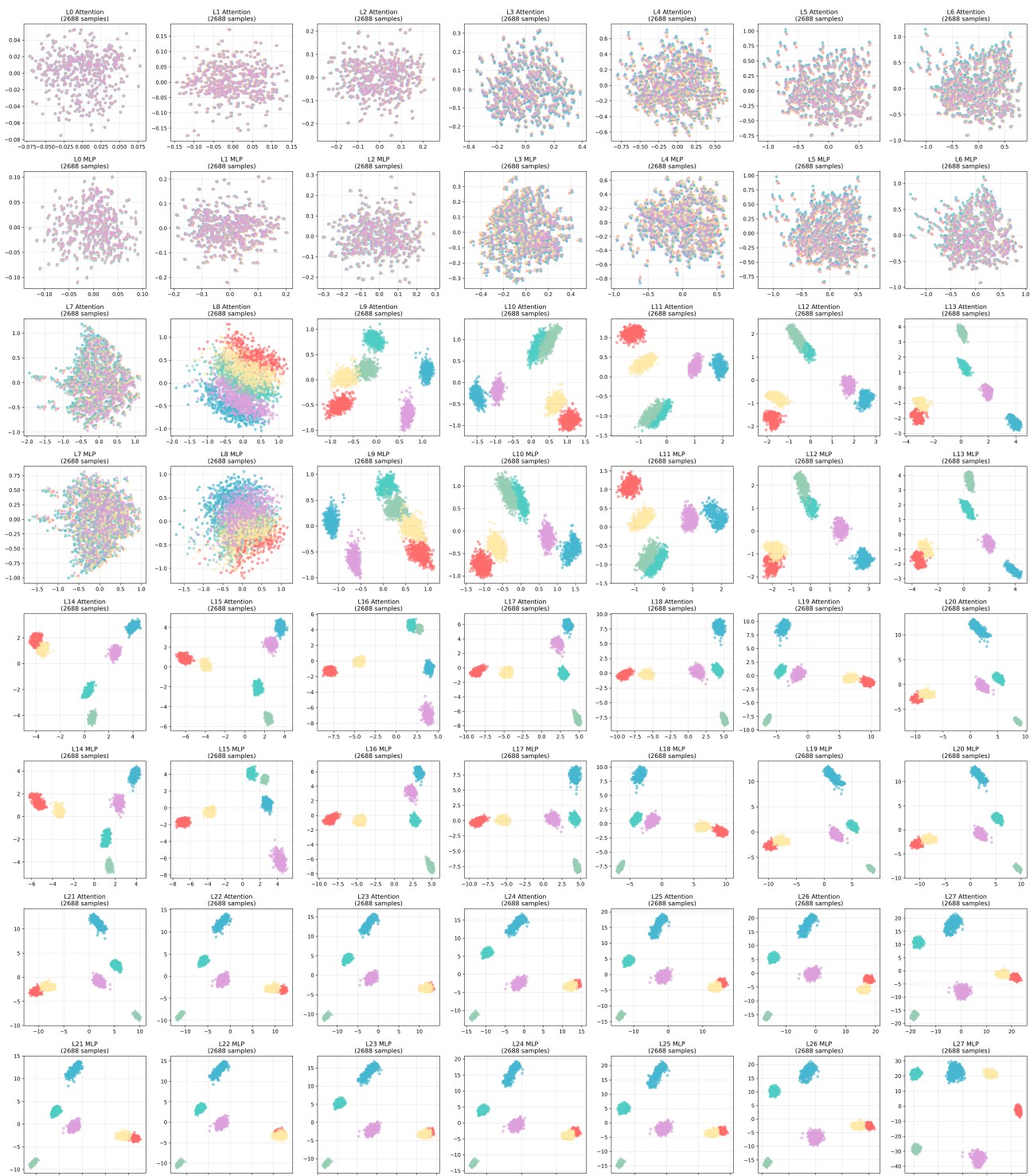

*Figure 4.* The layer-wise clustering visualizations of hidden states for all samples successfully guided by the prompting-based method.

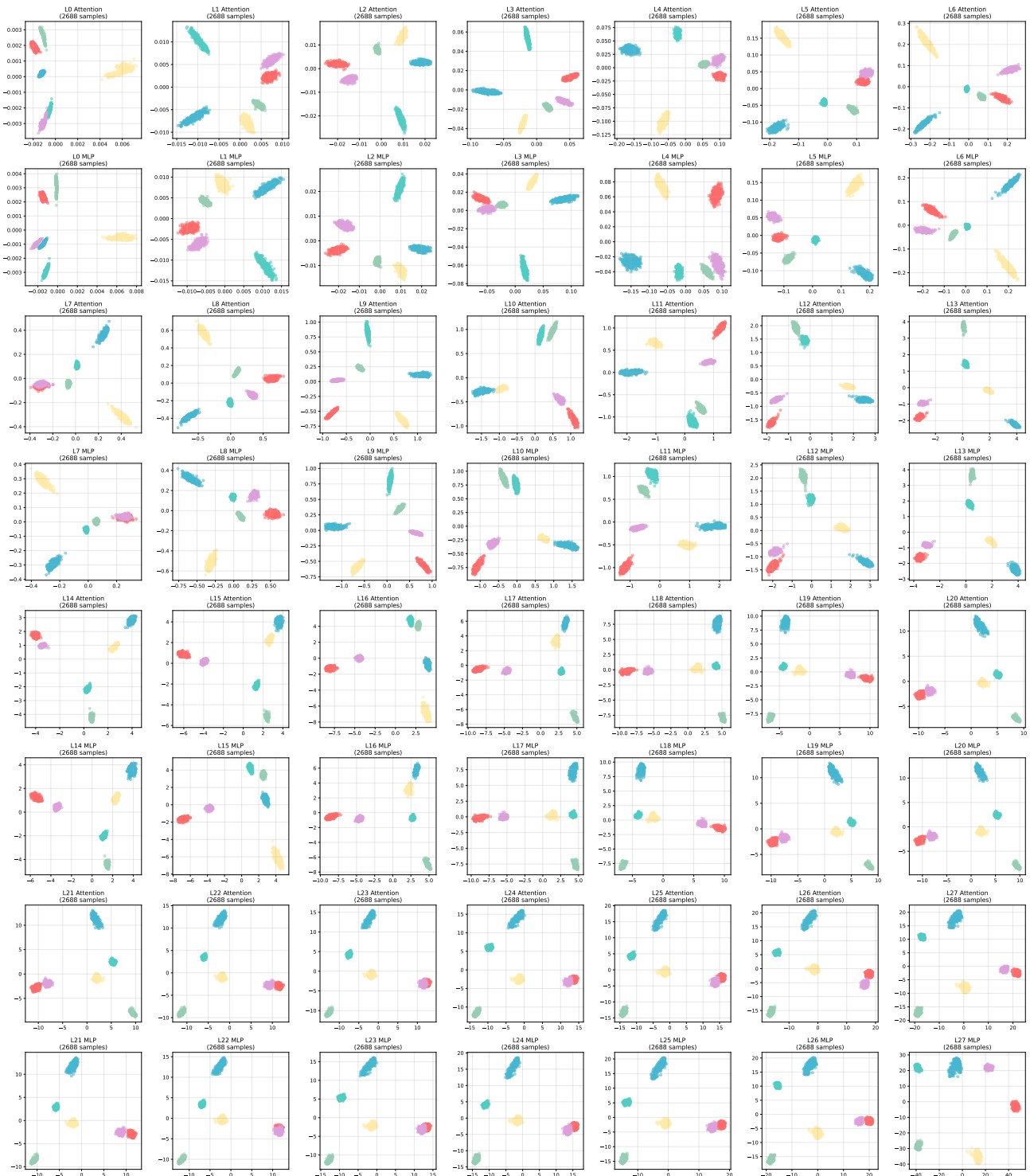

*Figure 5.* The layer-wise clustering visualizations of pure emotion vectors.

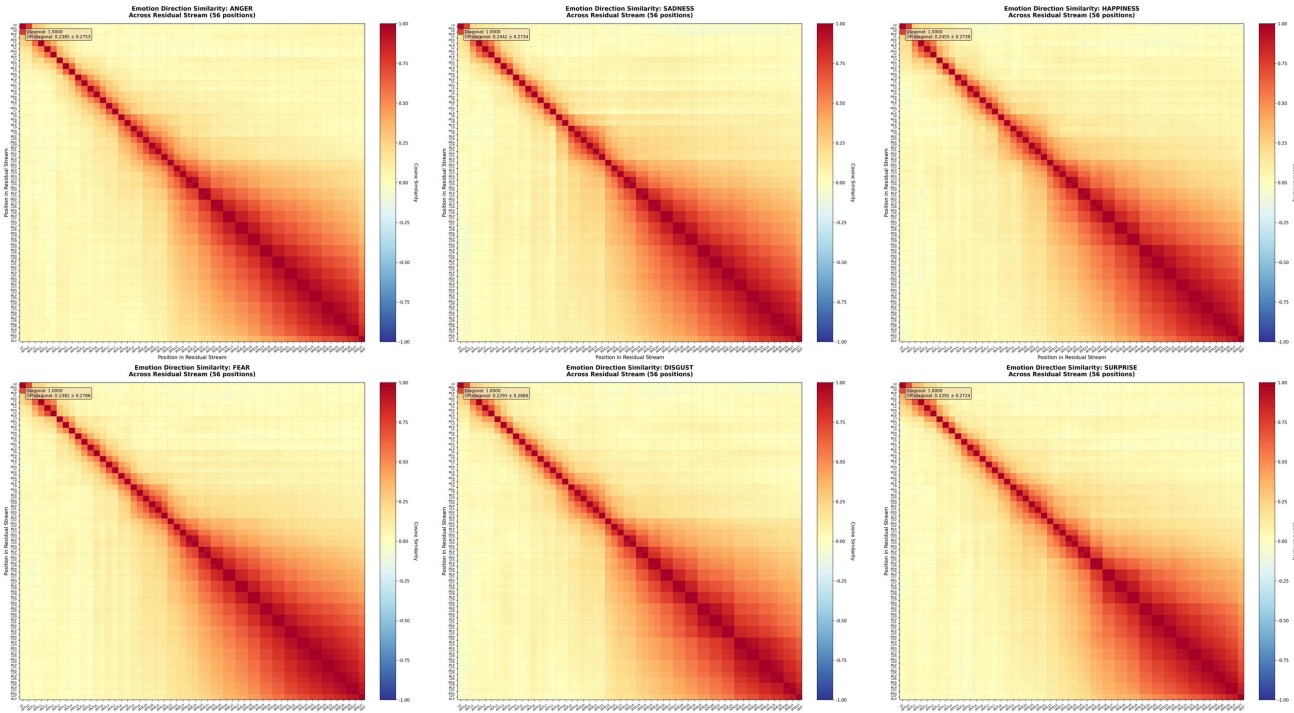

*Figure 6.* Cosine-similarity heatmaps of emotion directions across all 56 residual positions for six emotions, showing stable high similarity (>0.90) in deeper layers (21–25).

## H. Experiments on Qwen2.5-7B-Instruct

*Table 3.* Emotion expression accuracy (%) of prompt-based and steering-based generation for all six emotions on **Qwen2.5-7B-Instruct**.

| Method | Anger | Sadness | Happiness | Fear | Surprise | Disgust |
|--------|-------|---------|-----------|------|----------|---------|
| Prompting (SEV) | 96.25 | 84.38 | 88.54 | 91.46 | 90.42 | 84.79 |
| Prompting (Test Set) | 95.83 | 87.29 | 90.83 | 93.54 | 94.17 | 88.33 |
| Steering (Test Set) | 0.60 | 4.90 | 93.30 | 0.00 | 92.60 | 3.70 |

### H.1. Emotion Expression Accuracy of Prompting and Steering

Table 3 reports the emotion expression accuracy of prompt-based and steering-based methods. While prompting achieves consistently high accuracy across all emotions (84-96%), steering exhibits a notable pattern: it succeeds for positive emotions (happiness and surprise: >92%) but fails for negative emotions (anger, sadness, fear, and disgust: <5%). This suggests that Qwen2.5-7B-Instruct has strong inherent resistance to negative emotion steering, likely due to safety alignments that prevent the model from expressing harmful or negative emotional content through direct representational manipulation.

### H.2. Layer-wise Visualization of Emotion Representations

This section presents the layer-wise clustering visualizations of hidden states for all samples successfully guided by the prompting-based method on Qwen2.5-7B-Instruct, as well as pure emotion vectors' clustering visualizations (see Fig. 7 and Fig. 8).

### H.3. Results of MLP Neurons Intervention Experiments

This section presents the results of ablation & enhancement experiments conducted on Qwen2.5-7B-Instruct to verify the model-independent stability of our findings. The setup strictly follows the same procedures described in Section 6. As shown in Tables 4 and 5, Qwen exhibits the same long-tail pattern as LLaMA: emotion scores drop/increase sharply when a

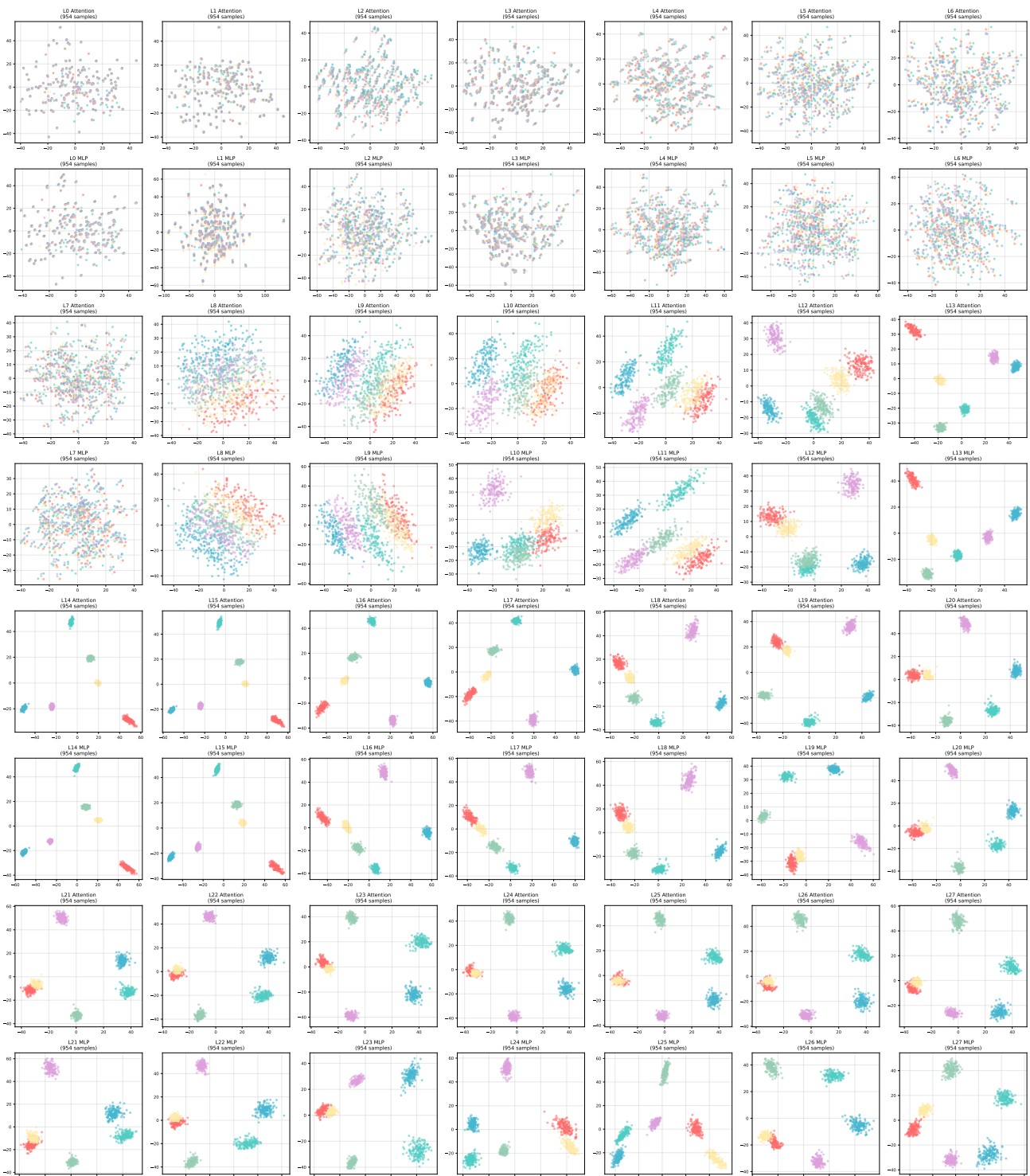

*Figure 7.* The layer-wise clustering visualizations of hidden states for all samples successfully guided by the prompting-based method on **Qwen2.5-7B-Instruct**.

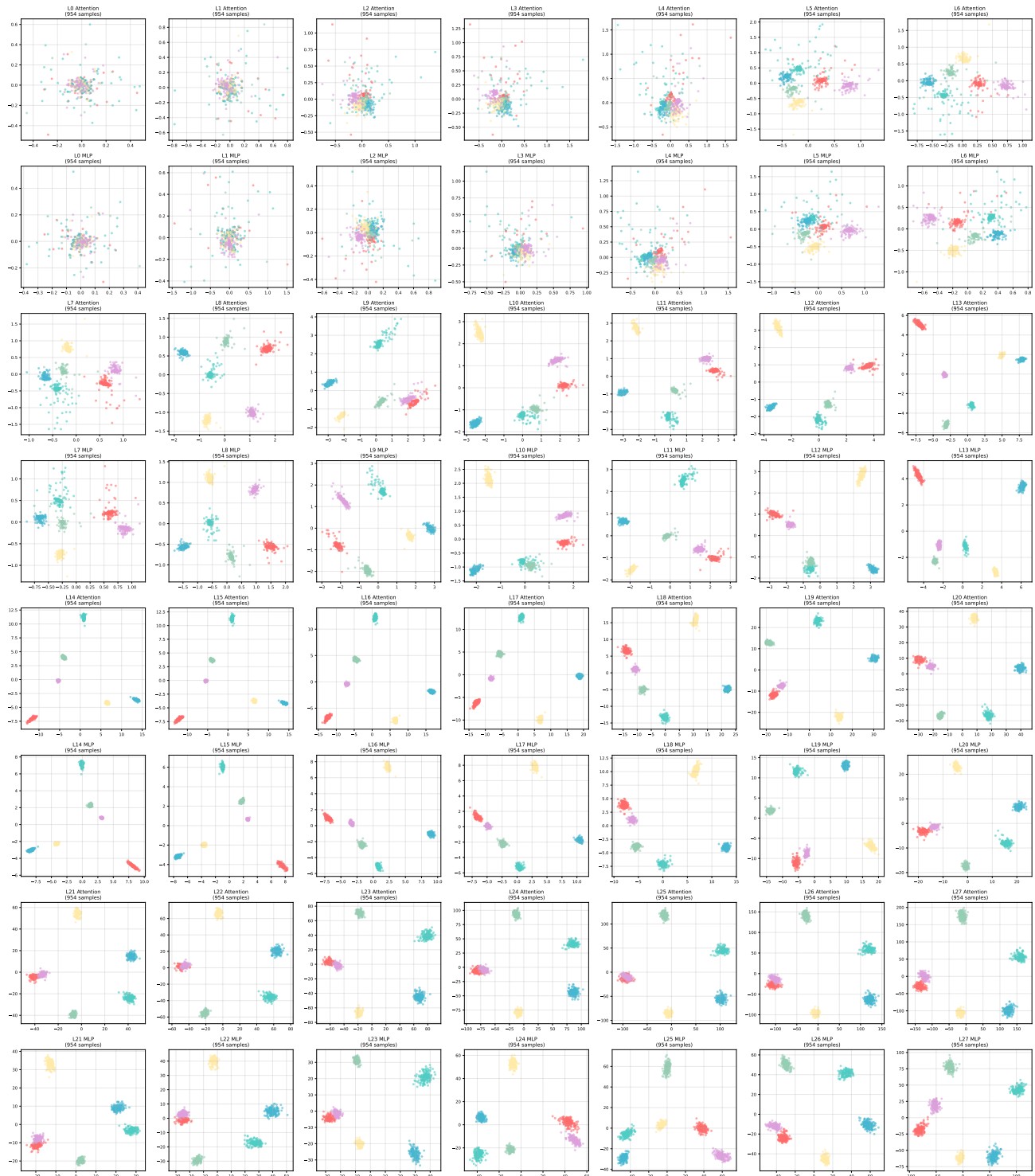

*Figure 8.* The layer-wise clustering visualizations of pure emotion vectors on **Qwen2.5-7B-Instruct**.

small number of top-ranked components are intervened, whereas intervening random components yields negligible change, confirming that our identified components really play an important role in forming emotion representations in LLMs, and only a few localized units dominate emotion expression.

*Table 4.* Ablation results on **Qwen2.5-7B-Instruct**. Emotion scores ($\Delta s$) decrease sharply when top-ranked neurons are ablated, while random ablations show minimal effect, mirroring the long-tail pattern observed in LLaMA.

| k | Anger | Sadness | Happiness | Fear | Surprise | Disgust | Random |
|---|-------|---------|-----------|------|----------|---------|--------|
| 0 | 0.00 | 0.00 | 0.00 | 0.00 | 0.00 | 0.00 | 0.00 |
| 2 | -100.30 | -155.54 | -153.47 | -71.40 | -178.11 | -199.91 | -0.27 |
| 4 | -171.15 | -221.37 | -260.03 | -131.05 | -245.09 | -254.98 | -0.38 |
| 8 | -296.61 | -283.06 | -338.65 | -197.56 | -340.23 | -332.96 | -0.76 |
| 16 | -397.25 | -353.57 | -402.78 | -286.53 | -408.74 | -405.73 | -1.30 |
| 32 | -497.72 | -439.28 | -473.85 | -386.88 | -522.58 | -509.94 | -3.13 |
| 64 | -566.12 | -519.96 | -552.92 | -460.57 | -609.85 | -602.93 | -5.05 |
| 128 | -657.05 | -606.55 | -635.91 | -548.90 | -674.13 | -668.64 | -9.81 |
| 256 | -749.72 | -691.59 | -714.90 | -621.62 | -767.92 | -752.66 | -21.15 |
| 512 | -836.63 | -795.71 | -795.64 | -740.43 | -833.84 | -839.28 | -41.72 |

*Table 5.* Enhancement experiment results on **Qwen2.5-7B-Instruct**: Impact of enhancing top-k MLP neurons on emotion intensity ($\Delta s$) for scale factor 1.0. Positive values indicate stronger emotion enhancement.

| k | Anger | Sadness | Happiness | Fear | Surprise | Disgust | Random |
|---|-------|---------|-----------|------|----------|---------|--------|
| 0 | 0.00 | 0.00 | 0.00 | 0.00 | 0.00 | 0.00 | 0.00 |
| 2 | 27.66 | 47.72 | 56.84 | 28.07 | 63.51 | 81.68 | 0.07 |
| 4 | 49.07 | 74.13 | 120.42 | 43.65 | 93.25 | 103.95 | 0.08 |
| 8 | 79.36 | 95.87 | 120.68 | 57.48 | 135.66 | 122.03 | 0.24 |
| 16 | 109.25 | 133.57 | 145.88 | 88.77 | 183.86 | 152.59 | 0.34 |
| 32 | 155.65 | 169.27 | 176.17 | 103.94 | 229.29 | 201.22 | 0.71 |
| 64 | 194.43 | 203.02 | 198.51 | 138.36 | 271.75 | 232.80 | 1.29 |
| 128 | 230.85 | 225.38 | 224.84 | 162.71 | 302.17 | 262.59 | 2.15 |
| 256 | 248.49 | 246.82 | 234.40 | 193.01 | 320.71 | 284.72 | 4.10 |
| 512 | 269.03 | 271.94 | 255.48 | 226.23 | 343.89 | 304.38 | 7.84 |

## H.4. Results of Attention Head Intervention Experiments

This section reports the results of attention head-level intervention experiments on Qwen2.5-7B-Instruct, examining the causal role of attention heads in emotion expression through both enhancement and ablation approaches.

Table 6 presents enhancement results that top-k attention heads are enhanced to amplify emotion intensity, while Table 7 shows masking results where top-k heads are ablated by zeroing their outputs. The positive $\Delta s$ values in enhancement and negative values in masking provide converging evidence that the identified attention heads are causally important for emotion expression.

*Table 6.* Attention head enhancement experiment results on **Qwen2.5-7B-Instruct**: Impact of enhancing top-k attention head on emotion intensity ($\Delta s$) for scale factor 1.0. Positive values indicate stronger emotion enhancement through head-level intervention.

| k | Anger | Sadness | Happiness | Fear | Surprise | Disgust | Random |
|---|-------|---------|-----------|------|----------|---------|--------|
| 0 | 0.00 | 0.00 | 0.00 | 0.00 | 0.00 | 0.00 | 0.00 |
| 1 | 42.14 | 27.99 | 34.25 | 21.66 | 112.00 | 17.13 | 1.17 |
| 2 | 88.41 | 55.71 | 41.74 | 40.82 | 157.14 | 48.76 | 6.69 |
| 3 | 141.91 | 77.07 | 66.54 | 52.29 | 224.57 | 85.62 | 19.64 |
| 4 | 165.92 | 92.00 | 105.04 | 59.10 | 244.20 | 100.10 | 14.18 |
| 5 | 183.27 | 96.75 | 115.89 | 76.69 | 275.41 | 109.23 | 19.06 |
| 6 | 191.34 | 104.51 | 122.30 | 97.59 | 291.51 | 116.12 | 22.53 |
| 7 | 200.64 | 109.28 | 133.08 | 98.24 | 306.98 | 128.96 | 33.03 |
| 8 | 211.47 | 111.87 | 134.33 | 96.04 | 300.42 | 136.32 | 22.42 |
| 9 | 215.03 | 109.82 | 125.16 | 100.66 | 287.39 | 148.22 | 43.12 |
| 10 | 212.95 | 115.11 | 135.68 | 105.07 | 305.84 | 151.16 | 42.86 |

*Table 7.* Attention head ablation experiment results on **Qwen2.5-7B-Instruct**: Impact of masking top-k attention heads on emotion intensity ($\Delta s_{\mathrm{mean}}$). Negative values indicate emotion reduction after head masking.

| k | Anger | Sadness | Happiness | Fear | Surprise | Disgust | Random |
|---|-------|---------|-----------|------|----------|---------|--------|
| 0 | 0.00 | 0.00 | 0.00 | 0.00 | 0.00 | 0.00 | 0.00 |
| 1 | -119.02 | -62.08 | -78.53 | -120.64 | -153.41 | -109.34 | -128.53 |
| 2 | -225.48 | -170.65 | -150.63 | -216.03 | -219.96 | -223.36 | -170.60 |
| 3 | -286.47 | -231.54 | -185.87 | -268.63 | -269.73 | -286.77 | -240.39 |
| 4 | -325.84 | -309.07 | -241.42 | -319.77 | -320.56 | -355.15 | -209.97 |
| 5 | -358.13 | -320.20 | -260.47 | -350.51 | -349.54 | -377.35 | -195.45 |
| 6 | -391.35 | -362.16 | -313.11 | -368.56 | -371.71 | -409.78 | -252.30 |
| 7 | -454.21 | -419.93 | -330.56 | -434.26 | -426.61 | -474.55 | -274.80 |
| 8 | -544.44 | -501.72 | -433.99 | -514.20 | -517.43 | -562.08 | -322.77 |
| 9 | -466.15 | -439.11 | -358.29 | -443.03 | -451.28 | -491.12 | -348.99 |
| 10 | -478.86 | -459.43 | -375.23 | -477.13 | -469.62 | -515.29 | -357.83 |

## H.5. Emotion Expression Accuracy with Scale 2.0 Steering

This section reports the emotion expression accuracy of circuit-based steering with a scale factor of 2.0 on Qwen2.5-7B-Instruct across different valence contexts (positive, neutral, and negative). The results demonstrate how steering effectiveness varies across emotions and context valences. Detailed per-valence accuracy for all six emotions is summarized in Table 8, which surpasses the performance of the steering-based eliciting method on the test set.

*Table 8.* Emotion expression accuracy (%) of circuit-based steering (scale 2.0) across different valence contexts on **Qwen2.5-7B-Instruct**.

| Valence | Anger | Sadness | Happiness | Fear | Surprise | Disgust | Average |
|---------|-------|---------|-----------|------|----------|---------|---------|
| Positive | 7.50 | 31.87 | 100.00 | 31.87 | 100.00 | 100.00 | 61.87 |
| Neutral | 8.12 | 42.50 | 100.00 | 37.50 | 97.50 | 100.00 | 64.27 |
| Negative | 39.38 | 95.62 | 60.00 | 52.50 | 86.88 | 100.00 | 72.40 |
| Average | 18.33 | 56.66 | 86.67 | 40.62 | 94.79 | 100.00 | 66.18 |

