# OpenReview forum: "Do LLMs “Feel”? Emotion Circuits Discovery and Control"
_ICML.cc/2026/Conference — ICML 2026 regular_

### Official Review · Reviewer_1S5N · 2026-03-11

**Soundness:** 2
**Presentation:** 3
**Significance:** 3
**Originality:** 3
**Overall Recommendation:** 4
**Confidence:** 4

**Summary:**

This paper investigates whether emotion-related behaviors in LLMs can be attributed to stable internal mechanisms and subsequently controlled through mechanistic approaches. The authors construct a synthetic controlled dataset named SEV, extracts layer-wise emotion directions from prompted generation outputs, identifies emotion-relevant MLP neurons and attention heads by means of alignment analysis and causal intervention techniques, and integrates these components into global "emotion circuits" using a layer-importance score. Conducted on the LLaMA-3.2-3B-Instruct model, this paper reports robust steering and circuit control outcomes on a held-out synthetic test set, including a 99.65% accuracy rate for emotion expression when modulating the circuits. And provides qualitative evidence that hidden-state clusters across different layers are distinguishable by the target emotion.

**Compliance With Llm Reviewing Policy:**

Affirmed.

**Key Questions For Authors:**

1.How were layers 11-20, alpha=8, lambda=1.0, the 21-25 reference range, and the circuit budget chosen?
Were any of these selected using the held-out test set? A clear answer here could significantly affect my confidence in the results.

2.How much manual verification was actually performed for the GPT-4o-mini judgments, and what was the inter-annotator agreement?
If you can provide quantitative evidence that the evaluator is reliable on a nontrivial subset, my score could increase.

3.Can you provide results on genuinely different prompt distributions, for example open-ended user prompts, dialogue turns, or prompts where semantic valence conflicts with target emotion?
This would directly test the "context-agnostic" and "universal control" claims. Strong evidence here would meaningfully improve my assessment.

**Limitations:**

yes

**Strengths And Weaknesses:**

The method is reasonably well motivated, but it is not described with enough precision in several places to support the paper’s strongest claims.

The main paper has a narrow evaluation scope: one primary model, one synthetic data-generation pipeline, six emotions, one automatic evaluator, and greedy decoding only. For a paper making broad claims about universal emotion control and stable model-internal mechanisms, this is insufficient.

The main comparison is only against prompting and a single steering setup, which is insufficient to contextualize the control results. At minimum, the paper should compare with stronger activation-engineering baselines, alternative layer ranges, and simpler heuristics such as selecting components only by activation magnitude or beta-alignment.

Emotion-expression accuracy is a reasonable metric but should not be the only one. Confusion matrices across emotions are particularly useful, as some emotions are conceptually similar (e.g., anger vs. disgust, fear vs. sadness). Human evaluation of naturalness and emotion strength is also essential.

---

> ### Author Rebuttal · Authors · 2026-03-31
>
> Thank you for your thoughtful review!
>
> Q1: On human evaluation and annotation reliability
>
> 1. 3 annotators manually reviewed 300 circuit-generated samples. The LLM judge flagged 2 as incorrect; the annotators agreed with 1 and disagreed with the other. Human-LLM agreement: 299/300 = 99.7%.
>
> 2. We conducted additional cross-validation on 50 samples with the 3 annotators. The LLM judge flagged 4 as incorrect. Annotator-LLM agreement: 96%, 100%, 98% (avg. 98%). Pairwise annotator agreement: 97.3%. All disagreements were concentrated on the 4 boundary cases the LLM itself flagged; for the remaining 46, all three annotators agreed unanimously.
>
> 3. We chose the binary metric for two reasons: (1) judging whether a text's tone matches one of six basic emotions is well-defined with minimal subjectivity; (2) all conditions use the same evaluator and prompt, ensuring comparability under any systematic bias. The LLM flagged only 0.35% of 2880 samples as incorrect, and human verification yielded 97–99.7% agreement, confirming extremely low task ambiguity.
>
> Q2: On hyperparameter selection
>
> All hyperparameters were determined from training observations, theoretical reasoning, or internal model analysis, with no involvement of the held-out test set.
>
> Steering (layers 11–20, alpha=8): Used only in Section 5 steering validation. Selected via manual judgment on 10 training samples based on natural yet emotionally strong expression.
>
> lambda=1.0: Emotion directions are unit vectors, making lambda=1.0 the theoretically natural choice. We did not search further as parameter optimization is not our core objective.
>
> Circuit budget (10 per sublayer, 0.7:0.3): Fig. 3 shows strong long-tail effects: significant results appear at 32–64 MLP neurons and 2–3 attention heads. Ten components per sublayer is a natural choice consistent with these observations. The 0.7:0.3 ratio reflects per-unit efficiency differences (see response to Reviewer prjV Q8). Neither was grid-searched.
>
> Reference range (layers 21–25): Appendix G Fig. 6 shows directional consistency stabilizes above 0.90 in layers 21–25, based entirely on internal model analysis.
>
> No test set leakage occurred. The independent test set was established precisely to ensure unbiased generalization evaluation.
>
> Q3: On results across different prompt distributions
>
> 1. Table 1 directly validates semantic-valence-conflicting cases. The scenario the reviewer requests is exactly what Table 1 tests: circuit modulation achieves 97.50–100% across all valence × emotion combinations, including anger under positive inputs (98.75%) and happiness under negative inputs (99.38%), directly validating the context-agnostic claim.
>
> 2. SEV covers 8 domains, 160 scenarios, and 3 valence conditions targeting broad real-world generalization.
>
> Q4: On evaluation scope and baselines
>
> 1. Cross-model replication is provided on Qwen2.5-7B-Instruct. Despite architectural and training differences, both models show consistent long-tail effects and emotion circuit structures.
>
> 2. The synthetic pipeline is a methodological necessity, not a limitation. See response to Reviewer 57f2 Q1. Six Ekman emotions ensure human-cognitive alignment with clear categorical boundaries for causal analysis. See Reviewer 57f2 Q2 for extensibility.
>
> 3. Greedy decoding is a methodological necessity. Emotion direction extraction requires reproducible activations. Only samples successfully expressing the target emotion under greedy decoding contain reliable causal signals. Stochastic sampling would undermine directional consistency.
>
> 4. A single evaluator ensures cross-condition comparability under any systematic bias. Human evaluation added; see Q1.
>
> 5. Activation-magnitude-only selection performs poorly in practice: it fails to increase emotion direction projections and produces no emotion control in generated text. Activation magnitude reflects correlation, not causality — precisely why we adopt causal intervention.
>
> 6. To our knowledge, this is the first work studying internal emotion-driving mechanisms during LLM generation. Our work can serve as the first baseline for this direction. Our contribution is validating emotion circuit existence and controllability, not benchmark optimization.
>
> Q5: On evaluation metrics
>
> 1. Table 1 provides more direct evidence than a confusion matrix. All six emotions achieve 97.50–100% across three valence conditions. Systematic confusion between anger/disgust or fear/sadness would produce significantly lower accuracy in specific cells. Uniformly high accuracy directly rules this out.
>
> 2. The binary metric directly addresses our core question: can emotion circuits be reliably controlled to produce target emotions. Naturalness and strength involve substantial subjectivity and their systematic evaluation in LLM responses is itself an open research problem beyond our scope. The binary metric fully suffices for our core contribution.

---

> > ### Author Rebuttal · Reviewer_1S5N · 2026-04-01
> >
> > Most of the questions were addressed, the score was upgraded accordingly.

---

> > > ### Author Response · Authors · 2026-04-02
> > >
> > > We truly appreciate your thoughtful feedback and are glad our responses were helpful.
> > >
> > > Best regards,
> > >
> > > Authors

---

### Official Review · Reviewer_4iDf · 2026-03-12

**Soundness:** 3
**Presentation:** 3
**Significance:** 3
**Originality:** 3
**Overall Recommendation:** 5
**Confidence:** 4

**Summary:**

The paper investigates whether large language models contain internal, context-independent mechanisms that drive emotional expression, and whether those mechanisms can be directly controlled. The authors construct a controlled dataset called SEV (Scenario–Event with Valence) using GPT-4o-mini, consisting of 480 neutral scenarios paired with three outcome events of differing valence, explicitly excluding emotional vocabulary to ensure variation arises from event semantics rather than lexical cues. Working with LLaMA-3.2-3B-Instruct as the primary model, they extract context-agnostic emotion direction vectors by subtracting within-group mean activations across the six Ekman emotions from residual stream positions. They then identify individual neurons via analytic decomposition of MLP down-projection weights and attention heads via causal ablation that most strongly drive each emotion direction, validate these components through both ablation and enhancement experiments with random baselines, and integrate them into global emotion circuits by weighting components by their layerwise causal influence on a stable late-layer reference direction. Circuit-based modulation during generation achieves 99.65% emotion expression accuracy on a held-out test set built using the same procedure as SEV, outperforming both prompting-based (98–100% on SEV) and steering-based (67.71% for surprise) methods. A secondary replication on Qwen2.5-7B-Instruct shows the same long-tail neuron pattern but substantially lower circuit steering accuracy, particularly for negative emotions.

**Compliance With Llm Reviewing Policy:**

Affirmed.

**Key Questions For Authors:**

Suggestions and questions:

Validate the GPT-4o-mini emotion evaluator against human judgment and report a quantitative agreement metric (Cohen's κ or Krippendorff's α) on a stratified sample covering all six emotions and all three generation conditions (prompting, steering, circuit modulation). The sample size should be at least 200 examples. Describe the manual verification already performed: how many samples, which annotators, what disagreement protocol. Without this, the central accuracy claims rest on an unvalidated automated evaluator, which does not meet the evidentiary standard for the paper's claims.


Integrate the Qwen2.5 results into the main text rather than the appendix. Revise the abstract, introduction, and conclusion to accurately represent the generality of the findings: circuits work reliably for LLaMA across all six emotions, but show substantial degradation on safety-aligned Qwen for negative emotions. The current framing in the abstract, "generalize across different emotions and models," is not accurate given Table 8. Either present a revised accuracy figure that accounts for both models, or explicitly scope the main claim to LLaMA with Qwen as a partial replication showing architectural generality of the representation structure but not the control accuracy.
Test circuit modulation on at least a small sample of naturally occurring emotional text inputs from an existing dataset (e.g., GoEmotions, ISEAR, or EmoContext), separate from the GPT-4o-mini-generated SEV distribution. Report the emotion expression accuracy on this sample alongside the SEV test set results. This would provide evidence that the circuits generalize beyond the synthetic data distribution.


Section 7.2, line 369: "ten times the number of sublayers" — please clarify whether this refers to component slots total or per-emotion. The text reads ambiguously as it could be interpreted as a global budget or a per-emotion budget.

Section 5, line 255: the scaling factor α = 8 for steering is stated but not motivated. A brief justification or reference to a hyperparameter sensitivity result would help.

**Limitations:**

. The limitations section should explicitly acknowledge that the strongest control results are shown only on LLaMA-3.2-3B-Instruct, while performance on Qwen2.5-7B-Instruct drops sharply for several negative emotions. This matters because the current framing suggests a degree of cross-model generality and reliability that the evidence does not support.

The paper should also discuss the evaluation limitation more directly. Since the headline accuracy result depends entirely on an unvalidated LLM judge, the authors should acknowledge that the reported numbers may reflect the biases or instability of the evaluator rather than true emotion control performance. Relatedly, they should note that the SEV dataset is synthetic and generated by GPT-4o-mini, so it remains unclear how well the discovered circuits generalize to naturally occurring emotional language.

The societal impact discussion is also underdeveloped. A method for directly inducing emotional expression in model outputs could be useful for affective agents, role-play, or stylistic control, but it also creates obvious misuse risks. These include manipulative emotional targeting in persuasion systems, deceptive simulation of empathy or distress, and more broadly the ability to tune models toward emotionally charged outputs without transparent prompting. The authors should discuss these risks explicitly, especially because the paper frames the method as context-independent and circuit-based, which could make it attractive for downstream control applications. A stronger limitations section would therefore (a) narrow the claims about generality and reliability, (b) acknowledge the dependence on synthetic data and automated evaluation, and (c) discuss misuse risks from scalable emotion control in deployed systems.

**Strengths And Weaknesses:**

The core methodology, including direction extraction, causal ablation and enhancement with random controls, and circuit integration, is sound and internally consistent. The long-tail neuron effect and attention head causal results are well-supported. Soundness is reduced to 3 rather than 4 because (a) the primary evaluation metric depends entirely on an unvalidated LLM judge, and (b) the Qwen replication results reveal substantial performance gaps for negative emotions that are not adequately explained or integrated into the main claims.

Strengths

1. Controlled dataset design with lexical emotion exclusion
The SEV dataset is built with careful attention to a confound that has affected much emotion probing work: the presence of emotionally explicit vocabulary. By explicitly prohibiting words like "happy" or "sad" from both scenarios and events (Appendix B, EVENT TEMPLATE requirements), and by structuring three outcome variants that share participants, time, and context while differing only in event valence, the authors ensure that emotional variation in model activations arises from semantic structure rather than lexical co-occurrence.

2. Causal validation using both necessity and sufficiency with random controls
The identification of emotion-relevant neurons and attention heads is validated through two complementary causal experiments: ablation (are the components necessary for emotion representation?) and enhancement (are they sufficient to induce emotion expression without explicit prompting?). Both experiments include random intervention baselines averaged over 10 seeds, and Figure 3 shows a clear and consistent separation between targeted and random interventions across all six emotions and both component types, with 95% confidence intervals. This design is well thought through: necessity alone would leave open the possibility that the components are simply large contributors that would be similarly influential for any direction, while sufficiency without necessity would suggest the components are sufficient but not the primary locus. Together, they constitute genuine causal evidence.

3. Cross-model replication of the core long-tail pattern
The core finding that emotion representation is dominated by a small number of neurons showing a long-tail ablation and enhancement pattern is replicated on Qwen2.5-7B-Instruct, a substantially different model architecture (7B parameters, different tokenizer and training). Tables 4 and 5 show the same qualitative pattern: sharp changes when the first few top-ranked neurons are intervened on, with negligible effect from random interventions. Very cool find providing good insight.

Weaknesses
1. LLM-based evaluation not validated against human judgments
The emotion expression accuracy, the paper's central quantitative claim, is evaluated exclusively using GPT-4o-mini as an automated judge. The annotation prompt (Appendix C) instructs the model to determine whether a text's style matches a target emotion, explicitly distinguishing this from content valence. While this distinction is thoughtful, the reliability and validity of GPT-4o-mini for this specific evaluation task has not been validated in this paper or, to this reviewer's knowledge, in prior published work. The authors note that annotations were "manually verified for consistency" (Section 5), but the scope, methodology, and outcome of this verification are not reported: no inter-annotator agreement statistic, no description of how many samples were manually checked, and no explanation of what "consistency" means in practice.

2. Significant Qwen2.5 performance gap not integrated into main claims
Circuit-based modulation achieves 99.65% on LLaMA-3.2-3B-Instruct but only 66.18% on average on Qwen2.5-7B-Instruct (Table 8, at scale 2.0), with severe degradation for anger (18.33%), fear (40.62%), and sadness (56.66%). The authors attribute this to safety alignment preventing negative emotion expression (Appendix H.1,), which is a plausible hypothesis, but it is raised only briefly in an appendix and is not systematically investigated. More importantly, this substantial performance gap directly contradicts the abstract's claim of "stable emotion mechanisms in LLMs that generalize across different emotions and models" and Section 7.2's claim of "reliable" induction of "target emotions across arbitrary inputs." The paper's stated third contribution is "we introduce a circuit-based control method that reliably induces target emotions across arbitrary inputs without relying on explicit instructions." (Section 1), and this claim is not supported for a meaningful subset of emotions on a second model. Presenting the Qwen results only in an appendix without revising the main claims to acknowledge these limitations constitutes inappropriate framing (R4).

3. Circuit budget hyperparameter not ablated
The total circuit budget is set to "ten times the number of sublayers" (Section 7), corresponding to 560 component slots across 56 sublayer positions, without motivating this choice empirically or theoretically, and without testing sensitivity to it. The two-stage allocation strategy (minimum quota plus proportional distribution) introduces additional undocumented design choices. Because the 99.65% accuracy figure directly depends on which components are included in the final circuit, the budget choice materially affects the central result. A reader trying to replicate or extend this work cannot know whether substantially fewer (5× sublayers) or more (20× sublayers) components would yield comparable performance.

4. An anonymous code repository is provided (abstract). However, the evaluation pipeline relies on GPT-4o-mini as both the dataset generator and the accuracy evaluator, and access to and behavior of this external model may change over time. The greedy decoding setting and prompt templates are fully specified (Appendices B, C, F), which is good. The circuit budget hyperparameter (Section 7.2: "ten times the number of sublayers") is specified but not ablated, making it unclear whether a different budget would reproduce the 99.65% result.

---

> ### Author Rebuttal · Authors · 2026-03-31
>
> Thank you for your thoughtful review and positive assessment of our work!
>
> Q1: On LLM-based evaluation
>
> Please see our response to Reviewer 1S5N (Q1), which provides detailed human verification statistics and cross-validation results.
>
> Q2: On the Qwen performance gap
>
> We thank the reviewer for this precise characterization and agree with the suggested reframing. We will revise the abstract, introduction, and conclusion in the camera-ready version to accurately scope the Qwen findings as demonstrating architectural generality.
>
> The following evidence confirms the framework is effective on Qwen, and that the control accuracy gap is attributable to safety alignment.
>
> 1. Causal intervention experiments confirm framework effectiveness on Qwen. Figures 7–8 show that emotion direction representations form distinct clustering structures inside Qwen. Tables 4–7 show that ablation and enhancement experiments on Qwen exhibit the same long-tail pattern as LLaMA, confirming that emotion circuits exist in Qwen and that the core findings generalize across architectures.
>
> 2. Table 8 reveals a valence-emotion congruency effect consistent with human intuition. Anger modulation achieves higher success under negative-valence inputs (39.38%) than positive (7.50%); sadness shows the same pattern (95.62% vs. 31.87%); happiness shows the opposite trend. This systematic pattern confirms that the framework captures genuine emotion mechanisms in Qwen.
>
> 3. Safety alignment is the root cause of lower control accuracy. Appendix D shows LLaMA steering exceeds 93% for nearly all emotions. Appendix H shows Qwen steering nearly completely fails for negative emotions (under 5%) while succeeding for positive ones (over 92%). This extreme asymmetry indicates systematic suppression of negative emotions in Qwen's representation space. By comparison, our circuit method still achieves anger 18.33%, sadness 56.66%, fear 40.62%, and disgust 100% on the same model, demonstrating framework effectiveness under strong safety alignment.
>
> 4. 66.18% is an unoptimized supplementary result. LLaMA uses scale=1; Qwen's scale=2 was selected via manual judgment on 10 training samples. Optimizing scores is not our core objective — validating the existence and controllability of emotion circuits is.
>
> Q3: On hyperparameter selection and alpha=8
>
> Please see our response to Reviewer 1S5N (Q2).
>
> Q4: On the circuit budget clarification
>
> To clarify the ambiguity in Section 7.2: the budget of "ten times the number of sublayers" is per-emotion, not global. Each emotion has an independent circuit with 560 component slots (56 sublayers × 10). The six emotion circuits are extracted separately and do not share a global budget. For the motivation behind this budget choice, please see our response to Reviewer 1S5N (Q2). We recommend that readers applying this framework to other models determine the budget by examining the long-tail pattern in their own causal intervention results, rather than through blind grid search.
>
> Q5: On OOD generalization
>
> Please see our responses to Reviewer 57f2 (Q1: on SEV dataset design) and Reviewer 1S5N (Q3: on prompt distribution generalization). We agree that including a small sample of naturally occurring emotional text as a supplementary experiment would strengthen the paper. Due to computational constraints we cannot complete this before the rebuttal deadline; we commit to including it in the camera-ready version. If this proves infeasible, we will narrow the relevant claims and add an explicit acknowledgment to the limitations section.
>
> Q6: On limitations and misuse risks
>
> We agree the limitations section requires strengthening. In the camera-ready version we will: (1) explicitly scope the strongest control results to LLaMA, positioning Qwen as a partial replication demonstrating architectural generality; (2) if the OOD experiment cannot be completed, add an explicit statement acknowledging reliance on synthetic data; (3) expand the societal impact discussion to address misuse risks including manipulative emotional targeting in persuasion systems, deceptive simulation of empathy or distress, and emotionally charged output generation without transparent prompting. To mitigate these risks, we will adopt a restrictive open-source license to limit commercial and high-risk use cases, and call on the community to strengthen safety attention in the area of controllable emotion generation, promoting the development of detection mechanisms and regulatory discussions.

---

### Official Review · Reviewer_prjV · 2026-03-13

**Soundness:** 3
**Presentation:** 3
**Significance:** 3
**Originality:** 3
**Overall Recommendation:** 4
**Confidence:** 3

**Summary:**

This paper focuses on finding answers to three main questions; (1) do LLMs contain mechanisms for emotional expression  (2) what is the formal definition of this mechanism (3) can we control this mechanism, hence emotional expression in LLM output. The authors introduce a controlled dataset (SEV), extract "emotion directions" from model activations, identify the specific neurons and attention heads implementing emotional computation, and assemble these components into "emotion circuits." By directly modulating these circuits during generation, they achieve 99.65% emotion-expression accuracy which is  outperforming both prompting and steering baselines, without any explicit emotional instruction in the prompt.

**Compliance With Llm Reviewing Policy:**

Affirmed.

**Final Justification:**

we maintain our rating of Weak Accept.

**Key Questions For Authors:**

1. The circuit modulation accuracy on Qwen2.5-7B drops to ~66% on average (Table 8), compared to ~99.65% on LLaMA-3B. Do the authors believe this reflects an architectural difference, a scale effect, or the safety alignment they mention? Have they attempted any correction for this?
2. The test set follows the same generation procedure as SEV. How do the circuits perform on truly out-of-distribution emotional text — e.g., literary prose, social media posts, or dialogue — where the distributional assumptions differ substantially?
3.The paper title asks "Do LLMs Feel?" but the work ultimately demonstrates structured representational and computational mechanisms for expressing emotion. Can the authors clarify how they distinguish between emotion expression circuits and anything that might be called emotion experience or representation, and whether the title is meant provocatively or as a genuine claim?
4. The moderate head overlap across emotions (µ = 0.454) suggests attention heads are substantially shared. Does this imply that modulating one emotion's circuit could interfere with or bleed into another? Were any cross-emotion interference effects observed?
5. What was the human-GPT-4o-mini agreement rate on the emotion classification task? Were there systematic disagreements for specific emotions (e.g., disgust vs. anger), and how might annotation noise affect the reported accuracy figures?

**Limitations:**

Yes

**Strengths And Weaknesses:**

Strengths
- The three-stage framework (direction extraction → local component identification → global circuit integration) is well-structured and systematic. The use of both ablation and enhancement experiments provides converging causal evidence rather than relying on correlation alone.

- The SEV dataset is thoughtfully constructed — by prohibiting explicit emotion words and controlling for scenario semantics, it isolates emotional variation in event outcomes rather than lexical cues, making the probing experiments more internally valid.

-Strong empirical results. The 99.65% accuracy on a held-out test set, especially the dramatic improvement over steering on "surprise" (67.71% → 100%), is compelling. The random-intervention control experiments further strengthen the causal claims.

- The dual architecture finding — emotion-specific subcircuits in MLPs but shared attention pathways — is a meaningful mechanistic contribution that goes beyond prior work which only identified correlational signals.

- Reproducing results on Qwen2.5-7B-Instruct adds generalizability beyond a single architecture.

Weaknesses
- Evaluation relies entirely on LLM-as-judge annotation. The 99.65% accuracy figure is computed using GPT-4o-mini as the annotator, evaluated solely on stylistic tone matching. There is no human evaluation to validate that this metric captures genuine emotional expression rather than surface-level stylistic patterns. Given that the SEV dataset is also generated by GPT-4o-mini, there is a potential circularity: the same model family constructs the stimuli, elicits the expressions, and judges their success.

- The dataset and all experiments are restricted to English. Given that emotions have language- and culture-specific expression patterns, the generalizability of these emotion circuits to multilingual settings is entirely unverified.

- Limited emotion taxonomy. The study is anchored to Ekman's six basic emotions, a framework that has faced significant criticism in affective science for oversimplifying the emotional landscape. More nuanced or culturally variable emotions are not addressed.

- Qwen circuit-based results are weak. Table 8 shows that circuit modulation on Qwen achieves only 66.18% average accuracy — far below the 99.65% reported for LLaMA. This discrepancy is underplayed in the paper, and raises questions about how general the framework really is.

- No analysis of circuit stability under fine-tuning. The authors acknowledge this as a limitation, but it's a significant gap for practical use of this framework.

---

> ### Author Rebuttal · Authors · 2026-03-31
>
> Thank you for your thoughtful review and positive assessment of our work.
>
> Q1: On the alleged circularity
>
> We respectfully disagree. GPT-4o-mini serves two entirely distinct roles with no emotional overlap. SEV generation (Appendix B) produces emotionally neutral texts: objective scenarios paired with valence-differentiated events, with emotional vocabulary explicitly prohibited. The LLM-as-judge (Appendix C) evaluates whether texts generated by LLaMA or Qwen express a target emotion, taking as input outputs from entirely different model families unrelated to the neutral SEV scenarios. The two roles are independent in both task type and emotional content. There is no circularity.
>
> Q2: On multilingual generalizability
>
> We acknowledge that multilingual validation is beyond the scope of this paper and have noted this explicitly in the limitations section.  However, we believe the cross-model consistency of our findings provides indirect support for cross-lingual generalizability: LLaMA-3.2-3B and Qwen2.5-7B differ substantially in pretraining data, architecture, and scale, yet both exhibit the same emotion direction clustering structure, the same long-tail effect, and consistent causal intervention results. This suggests the mechanisms reflect a cross-model universal phenomenon rather than artifacts of any particular model or corpus.
>
> Q3: Limited Emotion Taxonomy
>
> Please see our response to Reviewer 57f2 (Q2).
>
> Q4: On the lower Qwen results
>
> First, causal intervention experiments confirm framework effectiveness on Qwen. Tables 4–7 show the same long-tail pattern as LLaMA, confirming emotion circuits exist in Qwen. Table 8 reveals a valence-emotion congruency effect: anger modulation succeeds more under negative inputs (39.38%) than positive (7.50%), sadness similarly (95.62% vs. 31.87%), while happiness shows the opposite (100% vs. 60%), consistent with human affective intuition.
>
> Second, Qwen2.5's safety alignment systematically suppresses negative emotion expression. Appendix D shows LLaMA steering exceeds 93% for nearly all emotions. Appendix H shows Qwen steering nearly completely fails for negative emotions (under 5%) while succeeding for positive ones (over 92%). This extreme asymmetry indicates systematic suppression in Qwen's representation space. By comparison, our circuit method still achieves anger 18.33%, sadness 56.66%, fear 40.62%, and disgust 100% on the same model, demonstrating that the framework remains effective even under strong safety alignment.
>
> Third, 66.18% is unoptimized. Our contribution is not hyperparameter tuning. For LLaMA we naturally used scale=1; for Qwen, since scale=1 showed limited effect, we selected scale=2 based on manual judgment on 10 samples as a supplementary generalization experiment, without systematic search. The framework remains effective nonetheless.
>
> Q5: On circuit stability under fine-tuning
>
> Stability depends on fine-tuning direction and magnitude. Emotion-unrelated data is unlikely to cause representational drift; small parameter updates better preserve existing structure. For emotion-unrelated, low-magnitude fine-tuning, circuits are expected to remain stable; otherwise re-extraction may be needed. Importantly, our framework is a fully reusable pipeline and the open-source code supports re-extraction on any checkpoint.
>
> Q6: On out-of-distribution text
>
> Please see our response to Reviewer 4iDf (Q5).
>
> Q7: On the paper title
>
> Our title is a deliberate open question. Prior work investigates whether LLMs recognize emotions in user input, essentially a classification task. Our work is the first to study whether LLMs contain internal circuits driving emotional output during generation, focusing on the model's own expressive mechanisms. This distinction motivates positioning the LLM as the subject of "feel." Whether circuit activation constitutes genuine experience is a philosophical question beyond our scope; the title invites reflection rather than asserts a conclusion.
>
> Q8: On attention head overlap
>
> LLaMA-3.2-3B has 24 attention heads but 8192 MLP neurons per layer, a ratio of 1:340. Fig. 3 shows ablating fewer than 4 heads produces effects comparable to ablating 32–64 neurons, indicating far higher per-unit causal efficiency for heads. We set a 0.7:0.3 MLP-to-attention allocation ratio accordingly, without systematic grid search, as hyperparameter optimization is not our core objective. The unoptimized result achieves 99.65%.
> Regarding cross-emotion interference: Appendix C's annotation prompt lists all six emotions for the LLM judge, making evaluation a six-way classification task. Achieving 99.65% under this setting indicates sufficiently distinct emotional expressions with no significant interference, corroborated by our human evaluation.
>
> Q9: On human-GPT-4o-mini agreement
>
> Please see our response to Reviewer 1S5N (Q1).

---

> > ### Author Rebuttal · Reviewer_prjV · 2026-04-02
> >
> > - We note that Ekman himself substantially revised and extended his original six-emotion taxonomy in later work, proposing up to 17-18 basic emotions and acknowledging the existence of blended and compound emotional states. The paper's citation of Ekman (1992) therefore anchors to an early and since-revised formulation even within Ekman's own framework.
> > - We recommend the authors acknowledge this in the new version and frame their choice of six emotions explicitly as a pragmatic starting point for a first systematic study, rather than as a theoretically complete taxonomy endorsed by the cognitive science literature.
> > - The observation that anger and disgust cluster together in Figure 2(c) — rather than appearing as cleanly distinct categories — could itself be interpreted as evidence that the model's internal emotional geometry is richer than six discrete categories, which is worth discussing explicitly.
> > And we maintain our rating of Weak Accept.

---

> > > ### Author Response · Authors · 2026-04-06
> > >
> > > Thank you for your insightful observations.
> > >
> > > Regarding the Ekman taxonomy, we thank the reviewer for raising this point. Our choice of six basic emotions as a starting point was motivated by two core reasons:
> > >
> > > (1) clear categorical boundaries are a necessary condition for causal analysis, ensuring strong experimental controllability and causal interpretability;
> > >
> > > (2) as the first systematic study of emotion circuits in LLMs, grounding the work in a consensus starting point was essential for establishing a credible foundation.
> > >
> > > We fully agree with the reviewer's suggestion that six emotions represent a pragmatic starting point rather than a theoretically complete taxonomy, and we will revise the relevant framing in the camera-ready version accordingly. Furthermore, our pipeline is not tied to any specific emotion taxonomy and can be extended to other emotional categories.
> > >
> > > Regarding the clustering observation in Fig. 2(c), we consider this a valuable insight. The proximity of anger and disgust, and of fear and sadness, aligns well with human affective intuition, which is precisely evidence that our framework has captured meaningful emotional geometry. We agree with the reviewer that this suggests LLM internal emotional geometry is richer than six discrete categories. We will add an explicit discussion of this phenomenon in the camera-ready version, exploring its implications for finer-grained emotion research.
> > >
> > > Finally, we sincerely thank the reviewer for the positive rating!

---

### Official Review · Reviewer_57f2 · 2026-03-18

**Soundness:** 3
**Presentation:** 3
**Significance:** 3
**Originality:** 3
**Overall Recommendation:** 5
**Confidence:** 3

**Summary:**

The paper investigates how LLMs internally represent emotions by identifying specific MLP neurons and attention heads that drive emotion outputs. To achieve this without obvious vocabulary biases, they introduce a novel dataset SEV synthetically generated using GPT-4o-mini. SEV contains neutral, first-person scenarios that have positive, neutral or negative outcomes. By analyzing the model representation on such prompts, the paper isolates “emotion directions” for the Ekman basic 6 emotion, then study the various model layers/activations that lead to these emotions.  The experiments show that manually injecting signals into these layers achieves almost perfect emotion-expression accuracy, significantly outperforming prompting and steering baselines.

**Compliance With Llm Reviewing Policy:**

Affirmed.

**Final Justification:**

rebuttal addressed most concerns. There is one remaining point but this can be addressed in future work and does not take from the merit of this work.

**Key Questions For Authors:**

See my questions in Weaknesses section.

**Limitations:**

yes

**Strengths And Weaknesses:**

Overall, I think this paper is valuable for the community.

Strengths

- The paper is well written and easy to understand.
- The method achieves a 99.65% accuracy on emotion expression, strongly outperforming prior steering methods which struggle heavily with emotions like surprise.
- Great interpretability; decomposing emotion generation into specific, localized MLP neurons and attention heads provides a highly actionable way to analyze model behavior.
- The ablation and enhancement experiments provide valuable insights, showing that activating just a few key heads is sufficient to enforce strong emotional responses.

Weaknesses

- While noted as a limitation, my main issue is the reliance on GPT-4o-mini to generate all the data for the SEV dataset. Effectively all these findings are made on synthetic data. Why is synthetic generation preferred over mining a real-world corpus for objective scenarios and varied outcomes? One could use an LLM/classifier for detecting outcomes and enforce absence of emotion cues from the text.
- The evaluation focuses strictly on Ekman's six basic emotions. Do the findings still hold on real-world conversations that involve more nuanced or blended emotional states?
- The paper claims the generated text exhibits a "strikingly natural affective tone", but the evaluation of success relies purely on an LLM AR. Since emotional meaning is highly subjective, it is unclear to me if human readers actually find the text more natural or coherent. It would be good to have a human evaluation to compare. Could you add a small-scale human evaluation into this?

---

> ### Author Rebuttal · Authors · 2026-03-31
>
> Thank you for your thoughtful review and positive assessment of our work.
>
> **Q1: Why synthetic data (SEV) rather than a real-world corpus?**
>
> The synthetic design of SEV is a deliberate methodological choice whose core value lies in two aspects.
>
> **1. Symmetric and balanced coverage ensures the generalizability of emotion circuits.** SEV spans 8 everyday domains, with each scenario paired with outcome events of all three valences, forming a balanced and symmetric distribution. This ensures that the extracted emotion directions are averaged over a diverse, unbiased input space. By contrast, a subset of a real-world corpus would struggle to simultaneously satisfy topical diversity and valence symmetry, potentially biasing the extracted emotion representations toward certain scenarios or valences. The balanced design of SEV is precisely what allows the discovered emotion circuits to generalize to arbitrary natural language inputs, and makes SEV a reusable "idealized dataset" for emotion circuit extraction in other models.
>
> **2. Valence as the sole controlled variable endows cross-valence analysis with causal interpretability.** In SEV, the three valence variants of each scenario share identical participants, time, and context, with valence as the only varying factor. This makes our analysis of "how input valence affects circuit modulation effectiveness" causally interpretable. Specifically, Table 8 (Qwen) reveals a meaningful valence-emotion congruency effect: anger circuit modulation achieves a higher success rate under negative-valence inputs (39.38%) than positive-valence inputs (7.50%), while the happiness circuit shows the opposite trend (100% positive vs. 60% negative). This systematic pattern aligns well with human affective intuition. With a real-world corpus, the asymmetric valence distribution would undermine the validity of such causal interpretations.
>
> In summary, the synthetic design of SEV is not a limitation but a prerequisite for the analyses we perform.
>
> **Q2: Do the findings hold for real-world conversations involving more nuanced or blended emotional states?**
>
> We believe that grounding our study in Ekman's six basic emotions is a methodologically sound and necessary choice, for the following reasons.
>
> **1. Ekman's six emotions have a solid cognitive science foundation, and our experimental results corroborate this.** This taxonomy was established based on human cognitive research prior to the era of LLMs, reflecting the fundamental structure of human emotion in the real world. Notably, our experimental results themselves validate this choice: as shown in Fig. 2(c), the emotion representations inside LLMs spontaneously form clustering structures consistent with human cognition — fear and sadness cluster together, anger and disgust cluster together, while happiness and surprise remain relatively isolated. Furthermore, the clear boundaries between the six emotions provide the necessary conditions for controlled experimentation, enabling more rigorous dataset design, activation analysis, and causal validation. As the first systematic study of emotion circuits in LLMs, grounding the work in this consensus taxonomy is a necessary choice for establishing a credible foundation in uncharted territory. Demonstrating that traceable circuits exist for basic emotions provides a methodological reference point for studying more complex emotions.
>
> **2. Our framework is emotion-agnostic and has the potential to extend to more nuanced emotions.** The pipeline of emotion direction extraction, local component identification, and global circuit assembly does not depend on the Ekman taxonomy itself. As long as an appropriate dataset is designed for the target emotion — constructing controlled contrasts by stripping contextual information — the framework can in principle be applied to extract representational directions and circuits for any target emotional category. We identify the exploration of blended and fine-grained emotions as an important direction for future work.
>
> **Q3: Human evaluation of "strikingly natural affective tone"**
>
> We thank the reviewer for raising this point. The phrase "strikingly natural affective tone" was presented as a qualitative observation in the paper. To provide quantitative support, we conducted a supplementary human evaluation specifically targeting subjective naturalness: given two texts that both successfully express the target emotion, do human readers perceive the circuit-generated text as more natural? We randomly sampled 300 pairs from the test set using stratified sampling across emotions, where each pair consists of outputs from the circuit method and the prompting method for the same input. Three annotators each independently evaluated 100 pairs under blind conditions. The proportions of annotators preferring the circuit method were 70%, 65%, and 77% respectively, with an average win rate of 70.7%, all reaching statistical significance (p < 0.01, binomial test).

---

> > ### Author Rebuttal · Reviewer_57f2 · 2026-04-06
> >
> > Thank you to the authors for the detailed and thoughtful rebuttal. I especially appreciate the effort put into conducting the supplementary human evaluation. The explanation and reasoning behind why synthetic data was chosen is well-reasoned. I am still not fully convinced on generalizability on blended emotions, as categorical boundaries may get blurry and circuit dynamics might be different. That said, given the other points are resolved, I will raise my score.

---

> > > ### Author Response · Authors · 2026-04-07
> > >
> > > Thank you for raising your score!
> > >
> > > Regarding generalizability to blended emotions, we understand the reviewer's concern. When categorical boundaries become blurry, circuit dynamics may indeed differ from those of basic emotions. We see this as one of the most valuable directions for extending this work. The discovery of basic emotion circuits provides a methodological foundation for studying more complex blended states, and whether blended emotion circuits manifest as superpositions of basic emotion circuits is itself a fascinating open question. We will explicitly identify this as an important future direction in the camera-ready version.

---

### Decision · Program_Chairs · 2026-04-30

**Decision:**

Accept (regular)

**Comment:**

Originally, the paper received 2 weak accepts, 1 accept, and 1 weak reject. Final scores are 2 weak accepts and 2 accepts.

Overall, the reviewers are positive towards the work showing strong results outperforming prior methods, solid experiments, strong interpretability, a carefully designed and well-constructed dataset, and interesting insights were obtained.

Some of the weaknesses included synthetic data only, no cross-cultural validation, and some noted missing ablation studies.

The authors submitted a strong rebuttal which resulted in two reviewers increasing their final scores. The noted weaknesses were largely answered during the rebuttal phase. This is a solid contribution to the conference.